# Chitin-based barrier immunity and its loss predated mucus-colonization by indigenous gut microbiota

Keisuke Nakashima [1], Satoshi Kimura[2,3], Yu Ogawa[2,8], Soichi Watanabe[4], Satoshi Soma[4,9], Toyoji Kaneko[4], Lixy Yamada[5], Hitoshi Sawada[5], Che-Huang Tung[6,10], Tsai-Ming Lu[6,11], Jr-Kai Yu [6], Alejandro Villar-Briones[7], Sakura Kikuchi[1] & Noriyuki Satoh [1]

Mammalian gut microbiota are integral to host health. However, how this association began remains unclear. We show that in basal chordates the gut space is radially compartmentalized into a luminal part where food microbes pass and an almost axenic peripheral part, defined by membranous delamination of the gut epithelium. While this membrane, framed with chitin nanofibers, structurally resembles invertebrate peritrophic membranes, proteome supports its affinity to mammalian mucus layers, where gut microbiota colonize. In ray-finned fish, intestines harbor indigenous microbes, but chitinous membranes segregate these luminal microbes from the surrounding mucus layer. These data suggest that chitin-based barrier immunity is an ancient system, the loss of which, at least in mammals, provided mucus layers as a novel niche for microbial colonization. These findings provide a missing link for intestinal immune systems in animals, revealing disparate mucosal environment in model organisms and highlighting the loss of a proven system as innovation.

[1] Marine Genomics Unit, Okinawa Institute of Science and Technology Graduate University, 1919-1 Tancha, Onna-son, Okinawa 904-0495, Japan. [2] Department of Biomaterials Science, Graduate School of Agricultural and Life Sciences, The University of Tokyo, 1-1-1 Yayoi, Bunkyo-ku, Tokyo 113-8657, Japan. [3] Department of Plant and Environmental New Resources, College of Life Sciences, Kyung Hee University, Seocheon-dong, Giheung-gu, Yongin-si, Gyeonggi-do 446-701, Republic of Korea. [4] Department of Aquatic Bioscience, Graduate School of Agricultural and Life Sciences, The University of Tokyo, 1-1-1 Yayoi, Bunkyo-ku, Tokyo 113-8657, Japan. [5] Sugashima Marine Biological Laboratory, Graduate School of Science, Nagoya University, 429-63 Sugashima, Toba 517-0004, Japan. [6] Institute of Cellular and Organismic Biology, Academia Sinica, 128 Academia Road, Section 2, Nankang, 115 Taipei, Taiwan. [7] Instrumental Analysis Section, Okinawa Institute of Science and Technology Graduate University, 1919-1 Tancha, Onna-son, Okinawa 904-0495, Japan. [8] Present address: Univ. Grenoble Alpes, CNRS CERMAV, 38000 Grenoble, France. [9] Present address: Research Center for Bioinformatics and Biosciences, National Research Institute of Fisheries Science, Japan Fisheries Research and Education Agency, 2-12-4 Fukuura, Kanazawa, Yokohama-shi 236-8648, Japan. [10] Present address: Department of Aquatic Biosciences, National Chiayi University, 300 University Road, 60004 Chiayi, Taiwan. [11] Present address: Marine Genomics Unit, Okinawa Institute of Science and Technology Graduate University, 1919-1 Tancha, Onna-son, Okinawa 904-0495, Japan. Correspondence and requests for materials should be addressed to K.N. (email: keisuke@oist.jp)

Mammalian guts harbor indigenous microbial communities that show high population densities, diverse taxonomic compositions, and beneficial effects on host health[1]. Mucosal immune systems maintain gut homeostasis by eliminating pathogens, while tolerating and harnessing the indigenous microbes for beneficial associations[2,3]. Reciprocally, gut microbiota affect proper development of mucosal immune systems by stimulating innate and adaptive immune responses[4,5]. Although it was generally believed that the memory competence of adaptive immunity enhances resistance to previously encountered pathogens, growing evidence suggests that it provides more versatile means to shape and manage a complex microbial community in the intestine[6,7]. In fact, gut microbiota of invertebrates, which lack adaptive immunity, are generally far less complex and prone to be shaped by environmental microbial composition[8,9]. It remains unclear how the mammalian gut microbiota arose and coevolved with mucosal immune systems in the diverse milieu of animal–microbe association[10]. Based on the notion that complex biological systems can be discriminated into ancestral and derived features when properly set in an evolutionary framework, we addressed these questions by conducting a comparative analysis of chordates, an animal lineage that includes two invertebrate groups, tunicates and lancelets, as well as vertebrates[11].

## Results

**Compartmentalization of the gut space by envelope membranes**. Chordates show a remarkable diversity of food habits that is accompanied by morphological changes in the pharyngeal region (Supplementary Fig. 1). We point out that the diverse food habit of chordates originated from a distinct type of particulate feeding. Tunicates and lancelets employ unique mucus nets secreted from the endostyle, a pharyngeal organ that is a chordate invention and that is homologous to the vertebrate thyroid, to separate particulate matter from seawater flowing through the gill slits[12] (Fig. 1a–d; Supplementary Fig. 2; Supplementary Movies 1 and 2). The high capacity and non-selectivity of this filtration system subject the intestinal mucosal surface to an immense and continual bacterial load, but how these invertebrate chordates protect themselves from food microbes that include potential pathogens is unknown. We found that the tunicate, *Ciona intestinalis* Type A, defecates filtrating mucus nets that are enveloped by transparent membranous structures (Fig. 1e, f). We observed by dissection that this envelope membrane first appears in the stomach and wraps mucus nets through the gut. Scanning electron microscopy (SEM) showed that the formation of envelope membranes proceed in the manner of delamination from the gut epithelium (Fig. 1g–k). Cross-sections revealed that envelope membranes confine ingested microbes to the luminal space, maintaining the ciliated epithelium free of microbes (Fig. 1m, n). PCR-amplification of 16S ribosomal RNA (rRNA) genes confirmed the axenic condition (Fig. 1o).

**Envelope membranes are framed with chitin nanofibers**. We then examined structural features and chemical composition of the envelope membranes. Alkaline removal of proteinous components from intact porous membranes revealed a multilayered, meshed framework of randomly oriented nanofibers (Fig. 1k, l). Nanofibers are the plausible morphology of natural chitin. The average size of the mesh was $65.6 \pm 23.0$ nm ($n = 100$), which is smaller than most marine bacteria (1 μm). Fourier transform infrared spectroscopy (FT-IR), which provides information of chemical composition, and X-ray diffraction, which provides scattering profiles of crystalline compounds, demonstrated that the purified frameworks are composed of chitin and cellulose

(Fig. 2a, b). Tunicates are the only animal group known to synthesize cellulose[13]. Negative-staining TEM confirmed two types of crystalline nanofiber: abundant thin fibers (>10 nm diameter) and sparse thick ones (>20 nm diameter) (Fig. 2c). Electron diffraction of a single thick fiber gave clear reflection signals characteristic of cellulose (Fig. 2d), whereas a bundle of thin fibers emitted obscured signals of chitin, probably due to the small crystallite size (Fig. 2e). Chemically purified frameworks, which can be visualized using fluorescent probes conjugated with chitin-binding domain protein (Fig. 2g), were eliminated by chitinase, but not cellulase treatment. In the chitinase reaction, mass spectrometry analysis detected the release of *N*-acetylglucosamine and *N*-acetylchitobiose, which are the expected degradation products of chitin (Fig. 2f, Supplementary Fig. 3). Together, these data show that the meshed framework of envelope membranes consists chiefly of chitin with intermingled cellulose nanofibers.

**Envelope membranes contribute to gut barrier immunity**. Next, we tested the possibility that the chitinous membranes are relevant to formation and maintenance of the axenic space over the gut epithelium (Fig. 1m–o). We isolated the *Ciona* chitin synthase gene, *Ci-CHS* (Supplementary Fig. 4), and verified its expression in the stomach and the intestine using RNA in situ hybridization (Fig. 2h). Inhibition of chitin synthase activity using a substrate analog Nikkomycin Z[14] caused disruption of envelope membranes, which allowed direct microbial contact with gut enterocytes (Fig. 2i). This caused a drop of survival rate from 76.2% ($n = 84$) in a control group, which was reared in filtered seawater, to 4.8% ($n = 84$) in the experimental group, reared in the presence of pathogenic marine bacteria (Fig. 2j). Because the antibiotic Streptomycin maintained higher survivorship (83.3%, $n = 102$), toxic effects of Nikkomycin Z to *Ciona* can be excluded, similar to the case of amphibians[14]. These data suggest that envelope membranes framed with endogenous chitin promote gut homeostasis by acting as a physical barrier.

**Proteome of envelope membranes**. The chitinous framework of envelope membranes is buried within the surface matrix (Fig. 1k, l). To gain functional insights into this matrix, we identified protein components of envelope membranes using mass spectrometry (MS)-based proteomic analyses (Supplementary Table 1). The most abundant protein component was Ci-MACPF1, a putative, secreted, pore-forming protein of the membrane-attack complex/perforin (MACPF) family[15] (Fig. 3a, Supplementary Fig. 5a–c). MACPF family proteins are essential for cytolytic activities in various organisms, e.g., nematocyst toxin, malaria virulence factor or human complement system[15]. Our attempt to assess recombinant Ci-MACPF1 proteins for cytolytic activities is in progress.

Second in abundance was a variable region-containing chitin-binding protein (VCBP) VCBP-C (Fig. 3b). This protein binds to gut luminal bacteria via its N-terminal variable-type immunoglobulin domains, thereby acting as an opsonin to enhance bacterial phagocytosis in the lamina propria[16]. It has been also suggested that the C-terminal chitin-binding domain (CBD) recognizes self and non-self chitin, based on immunostaining data for VCBP-C and a chimeric human IgG1 Fc-CBD protein[17]. Our proteomic data add another line of evidence in favor of this view. VCBP-C recognizes endogenous chitin in the envelope membranes. Furthermore, recombinant VCBP-C tethers gut-derived *Bacillus* sp. to chitin beads (Fig. 3c, Supplementary Fig. 5d). These data suggest that VCBP-C helps minimize microbial access to the epithelium by trapping bacteria on the chitinous barrier.

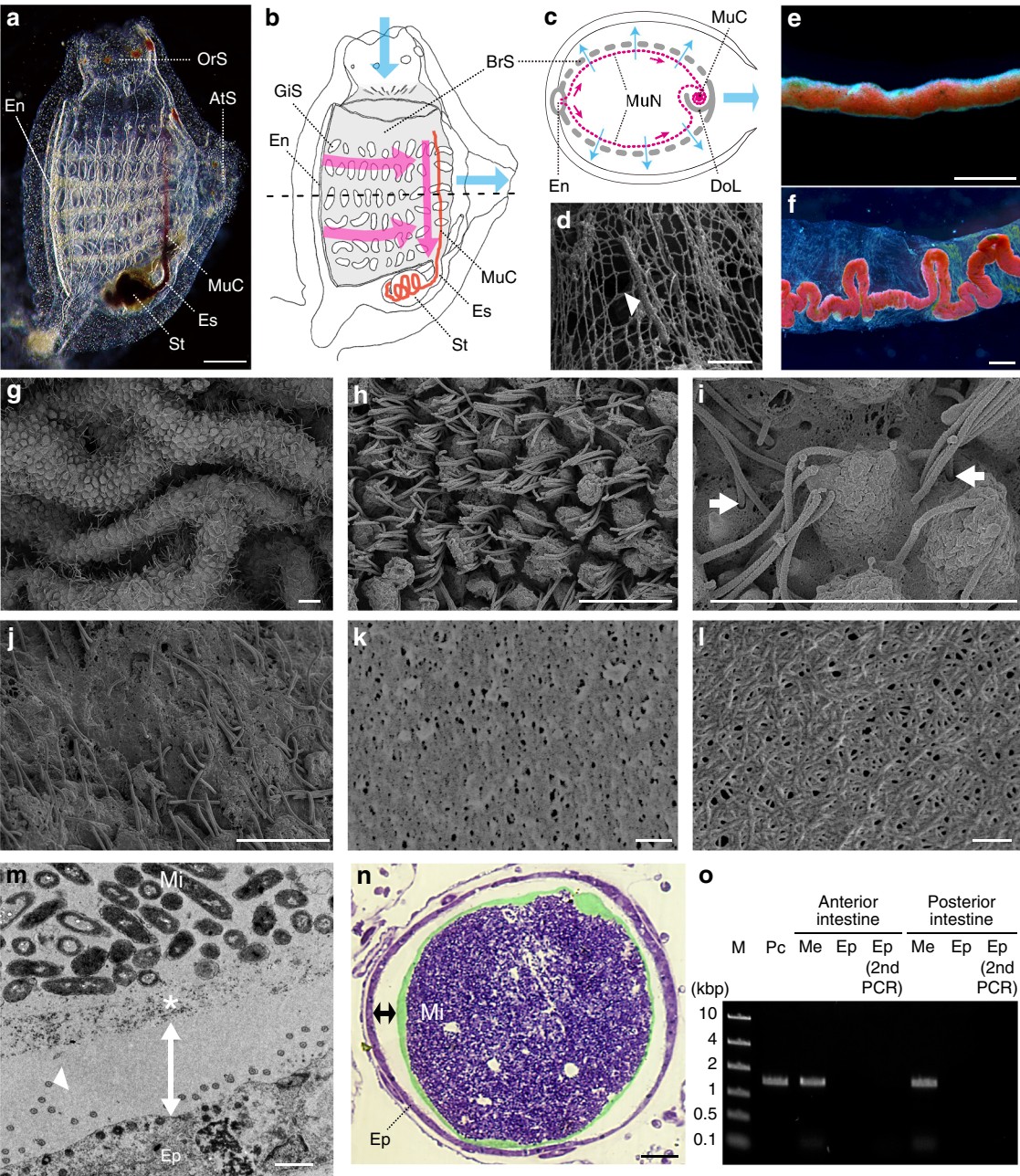

**Fig. 1** Gut barrier membrane of the tunicate *Ciona intestinalis* Type A. **a–c** The feeding mechanism of *C. intestinalis* Type A (**a**, juvenile specimen; **b**, anatomy; **c**, horizontal section at the dotted line in **b**). Cyan and magenta arrows denote the flows of seawater and mucus nets (MuN), respectively. While seawater, drawn from the oral siphon (OrS) into the branchial sac (BrS, gray), passes through the gill slits (GiS) to be expelled from the atrial siphon (AtS), particulate matter in seawater is trapped with mucus nets covering the inside of the branchial sac (magenta dotted lines in **c**). Mucus nets, secreted from the endostyle (En) and conveyed to the dorsal lamina (DoL), are rolled up as a single mucus cord (MuC, a red line in **b**), which is then transported posteriorly to the esophagus (Es) and the stomach (St). The mucus cord is recognizable due to trapped red beads. **d** An SEM image of rectangular mucus net and a trapped microbe (arrowhead). **e** A mucus cord isolated from the dorsal lamina. **f** Feces. A winding mucus cord is enveloped inside a transparent membrane. **g–l** The formation of envelope membranes (SEM images). **g** Intestinal mucosal surface with epithelial ridges. **h** The apical side of epithelial cells projects into the luminal space, and cilia extend from spaces between projections. **i** The epithelial surface is covered with a membrane that cilia penetrate (arrows). **j** A delaminating membrane from the epithelium. Cilia, but not projections, are recognizable. **k** The porous surface of a delaminated membrane. **l** The chemically purified framework of a membrane: meshed nanofibers. **m, n** Cross-sections showing axenic spaces (double-headed arrows) over the gut epithelium (Ep) (**m**, TEM; **n**, light microscopy). Intestinal microbes (Mi) are confined to the luminal space by multi-layered membranes (* or false-colored green). An arrowhead indicates one of the cilium sections. **o** Confirmation of the axenic condition by PCR amplifications of 16S rRNA genes. M, markers; Pc, positive control (food microbes); Me, isolated membranes enclosing food residues. Scale bars **a** 200 µm; **d, m** 1 µm; **e, f**, 500 µm; **g–j**, 5 µm; **k, l** 100 nm, and **n** 8 µm

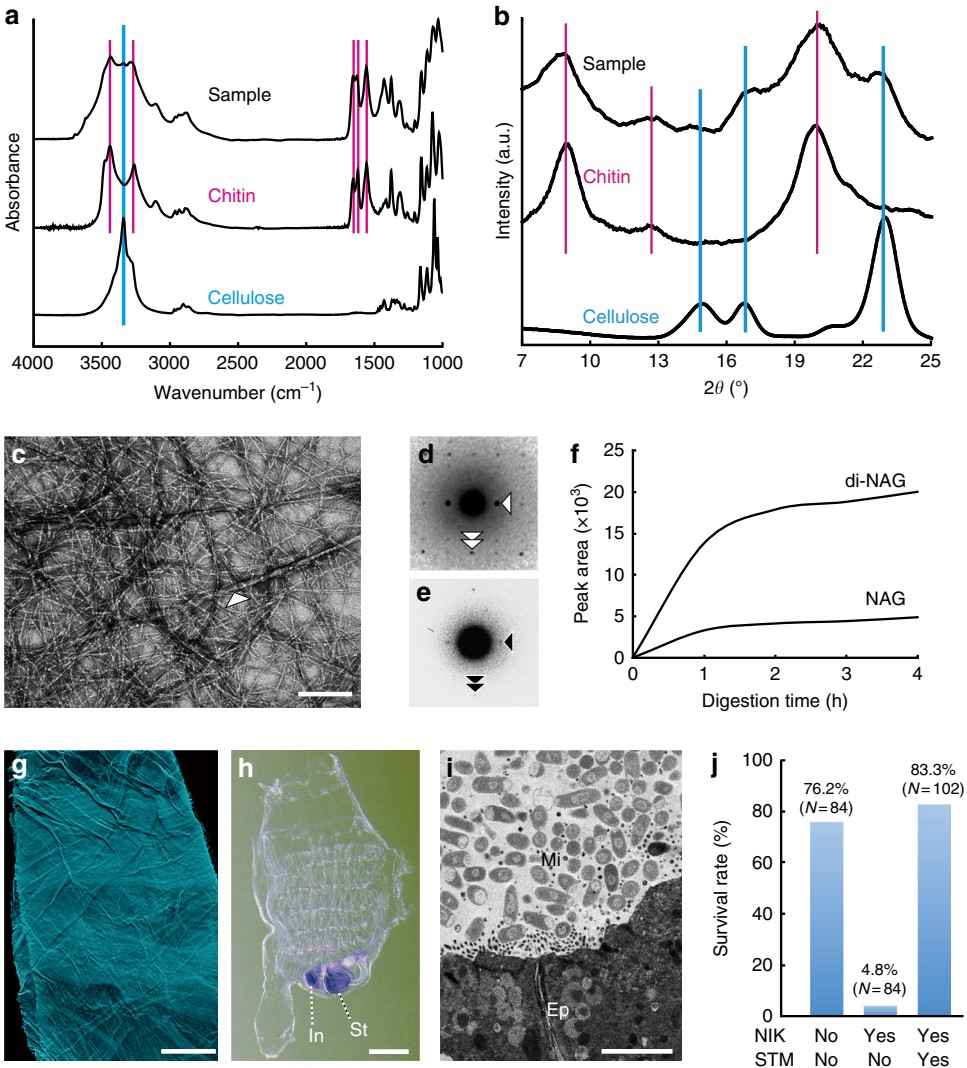

**Fig. 2** The chitinous framework of *Ciona* barrier membranes. **a**, **b** Spectroscopic and crystalline profiles of the meshed framework purified from *Ciona* barrier membranes (**a**, FT-IR spectra; **b**, X-ray diffractograms). Magenta and cyan vertical lines indicate peak positions that are specific to chitin (α allomorph) or cellulose (I_β allomorph), respectively. **c** A negative stain TEM image of a purified framework showing two types of crystalline nanofiber. A white arrowhead indicates a sparse, thick fiber meshed with abundant thin fibers. **d** An electron diffractogram of a single thick fiber showing cellulose (I_β allomorph)-specific reflections indexed with the lattice planes (110) (white arrowhead) and (004) (white double arrowhead). **e** An electron diffractogram of thin fibers, showing chitin-specific signals assigned to the lattice planes (020) (arrowhead) and (002) (double arrowhead). **f** Mass spectrometry-based time course relative quantification of *N*-acetylglucosamine (NAG) and *N*-acetylchitobiose (di-NAG) released from a purified framework under chitinase treatment. For details, see Supplementary Fig. 3. **g** A purified framework visualized with a fluorescent probe conjugated with a chitin-binding domain protein. **h** Whole-mount in situ hybridization showing expression of the chitin synthase gene *Ci-CHS* in the stomach (St) and anterior intestine (In). **i**, **j** Barrier membranes are essential for survival. **i** A TEM image of gut cross-section of the specimen treated with the chitin synthase inhibitor, Nikkomycin Z (30 μM). Intestinal microbes (Mi) directly contact the gut epithelium (Ep). **j** Reduced survivorship caused by Nikkomycin Z treatment, which can be compensated in aseptic conditions promoted by the antibiotic, streptomycin (50 μg/mL). NIK nikkomycin Z; STM streptomycin. Scale bars **c** 100 nm; **g**, **h** 100 μm, and **i** 5 μm

Third in abundance was a large mosaic protein (2880 amino acids) having 30 domains of 13 types. The overall arrangement of these domains and 113 cysteine residues is conserved with human gel-forming mucins (GFMs)[18] and von Willebrand factor (VWF)[19] (Fig. 3d, Supplementary Fig. 5e). GFMs and VWF diverge from a common progenitor gene by acquiring different modules in their central regions: densely *O*-glycosylated proline–threonine–serine-rich (PTS) domains in GFMs and platelet-binding VWA domains in VWF[20]. We noted a small PTS domain (50 amino acid residues) in the central region of the *Ciona* protein, so we named it Ci-GFM1. Ci-GFM1 retains all five cysteine residues that form intermolecular disulfide bonds essential for multimeric structures of GFM and VWF[18,19] (Fig. 3e,

Supplementary Fig. 5e). Ci-GFM1 also harbors a CBD. We thus predict that multimeric Ci-GFM1 lines the chitinous wall of the intestinal barrier, though this needs to be tested in future. Collectively, the proteomic data support the view that the intestinal physical barrier is immunologically fortified with matrix components (Fig. 4a).

**Intestinal chitinous membranes are prevalent in chordates**. The finding of chitin-based barrier immunity in the gut of the tunicate *Ciona* raises the question of how it is related to intestinal immune mechanisms of other animal groups (Fig. 4b). In many invertebrate groups, a membranous matrix surrounds food residues in

the midgut, persists through the intestine and often accumulates in fecal pellets[21]. This so-called peritrophic matrix (PM) contains chitin in arthropods and annelids, although other groups lack detectable chitin[21,22]. Because insect PMs are targets of pest control and malaria research[23], we were able to compare them in detail with the *Ciona* membrane. They share a mesh of chitin nanofibers synthesized by homologous chitin synthases, but they differ in protein composition. Insect PM proteomes commonly consist of proteins with multiple CBD or PTS domains, or both, but lack MACPF, VCBP, and GFM[23]. In contrast, our proteomic data suggest an affinity to mammals, because GFMs are the main structural components of mammalian mucus layers[18,24].

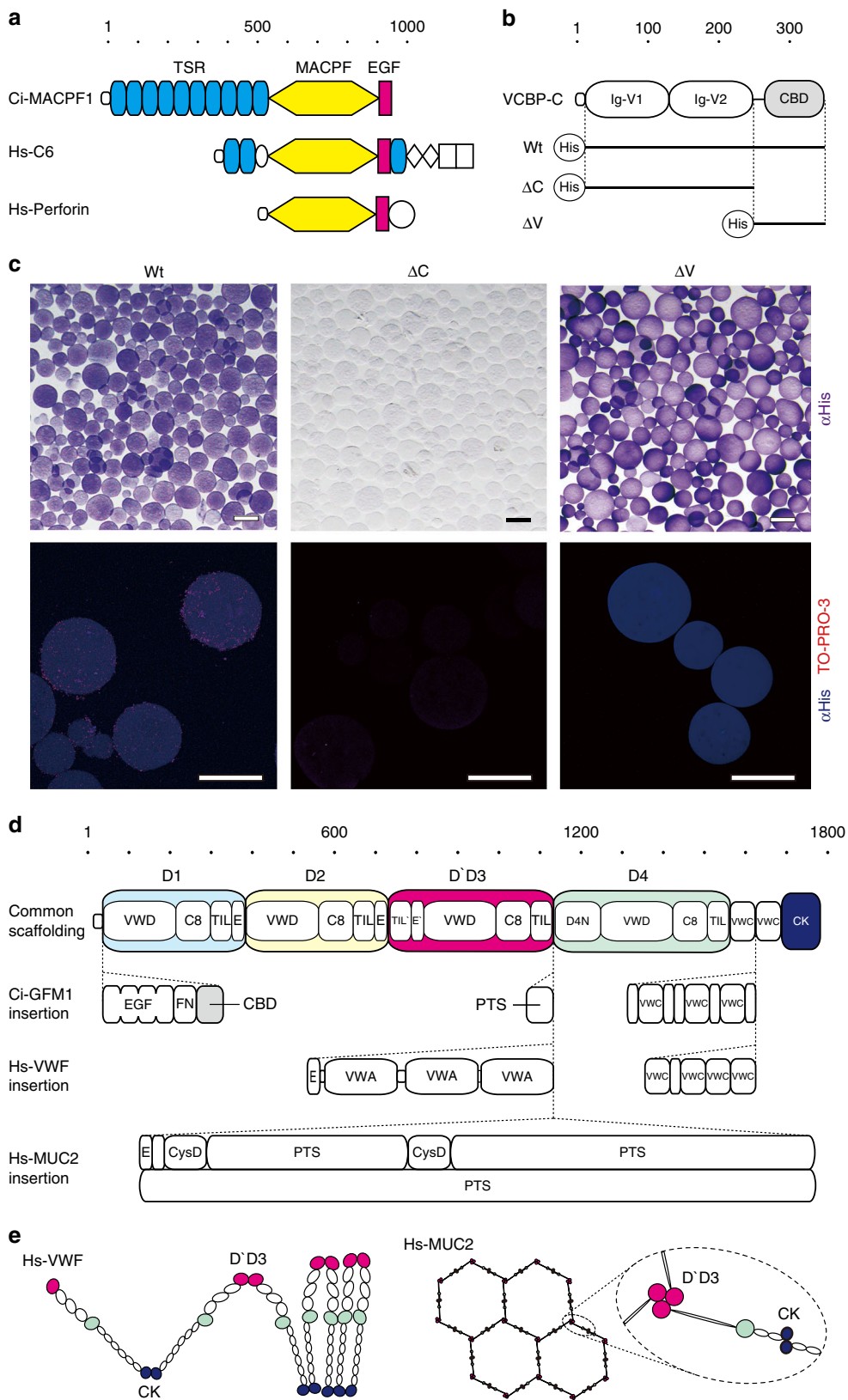

Therefore, we hypothesize that the envelope membrane of *Ciona* represents an evolutionary link between invertebrate PMs and mammalian mucus layers (Fig. 4b). To bridge the gap between them, we examined the guts of chordate lineages that occupy phylogenetically intervening positions between invertebrates and mammals, as follows: a basal chordate *Branchiostoma floridae* (lancelet), a jawless vertebrate *Eptatretus atami* (hagfish), and a jawed vertebrate *Oreochromis mossambicus*, known as Mozambique tilapia, one of the popular aquaculture species worldwide (ray-finned fish). Using structural, chemical, crystallographic, and gene expression criteria, we demonstrated intestinal chitinous membranes devoid of cellulose in these organisms (Figs. 5, 6b, Supplementary Fig. 6a–q).

**Chitin-based barrier immunity in ray-finned fish**. We further investigated the ray-finned fish, *O. mossambicus*, because the presence of intestinal chitinous membranes seemingly contradicts the widely accepted view that gut mucosal surfaces of ray-finned fishes are covered with a mucus layer colonized with indigenous microbes, as in mammals[25,26]. Actually, intestines of *O. mossambicus* harbor an indigenous microbial community, as revealed by 16S rRNA gene analysis of digesta (Fig. 6c and Supplementary Table 2). In the posterior intestine, called the distal major coil, the dominant group (61.8%) at the generic level was *Cetobacterium*, a fusobacterium that is widely distributed in intestines of freshwater fish and is regarded as the source of host vitamin B12[27,28]. The second most abundant group (5.1%) was the verrucomicrobe, *Akkermansia*, which includes a key mucin-degrader in human guts[29]. These indigenous microbes are associated with digesta mucus, which is derived from the gills and the esophagus[30] (Fig. 6d). Nevertheless, this microbial community was separated from the surrounding mucus layer that is secreted by goblet cells in the epithelial crypts, and enclosed by the chitinous membranes (Fig. 6e, f and Supplementary Fig. 6r–t). We noted a small number of DAPI signals at the surface of the mucus layer, yet it remains unclear whether these signals are occasional bacterial breaches or mucus-colonizing taxa. Numbers of nuclear-staining DAPI signals in the luminal or peripheral spaces, demarcated by a CBD-visualized membrane, were $95.9 \pm 29.0$ and $0.6 \pm 1.0$ (in a quadrat frame of $50 \times 50 \mu m$, $n = 26$), respectively.

This segregation between gut microbes and the mucus layer contrasted with what is known about mice, in which mucus organization varies along the longitudinal axis of the intestine[31]. While ileum mucus is loose and unattached to the gut epithelium (Fig. 6h), colonic mucus covers the epithelium, forming two

layers: the inner layer is firm and devoid of bacteria, and the outer is loose and densely colonized by gut microbes (Fig. 6i–l). Indeed, we were unable to detect chitin in mice by either CBD-staining or chemical purification. On the other hand, we obtained CBD-staining signals in gut sections of ray-finned fishes, zebrafish and rainbow trout (Fig. 6n–s). These data provide in situ profiles of possible chitinous membranes, irrespective of diverse gut morphology. We further attempted to confirm chitin by chemical purification from fish feces, but failed due to a paucity of chitin. Collectively, these data favor the view that the chitin-based ancestral system is somehow retained in ray-finned fishes, but was lost in lobe-finned fishes on the evolutionary course to mammals (Fig. 6m).

## Discussion

This comparative study showed the presence of chitin-based barrier immunity in chordate guts (Fig. 4a). While intestinal chitinous membranes, termed PM, have been appreciated for barrier immunity, nutrition and other physiological functions in invertebrates[21,32], it has long been held that chitin was lost in chordates[22]. This notion was recently challenged by mining chordate genomes for putative chitin synthase genes[33], followed by obtaining an infrared spectrum of chitin from Atlantic salmon scales[34]. Intestinal chitin has also been suggested in zebrafish[34] and *C. intestinalis* Type A[17] based on staining data using CBD and calcofluor-white, but it remains obscure because these molecules are not specific to chitin. For instance, this type of CBD, classified in the carbohydrate-binding module family 14, recognizes at least chitin, hyaluronan and N-glycans on glyco-proteins[35]. Calcofluor-white binds firmly to several β-1,3- and β-1,4-glucans beside chitin[36]. Given this technical limitation, care should be taken to avoid confusion due to misinterpretation of staining data, as exemplified by past cases for wheat germ agglutinin or aqueous iodine, known as chitosan test[21,22]. In these staining methods, the presence of chitin is sufficient to raise staining signals; however, staining signals does not necessarily mean the presence of chitin. Actually, these molecules are versatile and practical tools to detect "potential" chitin in situ (Fig. 6f, o, p, r, s). Instead, the present structural data at the nanoscale, combined with physical and chemical evaluations, demonstrated intestinal chitin in chordates and allowed us to consider its physiological relevance.

In light of animal phylogeny, the chitin-based barrier immunity in chordate guts bridges the gap between the invertebrate PM and the mammalian mucus layer, which have not been thought to

**Fig. 3** Protein components of the *Ciona* barrier membrane. **a** Domain structure of the membrane-attack complex/perforin family protein, Ci-MACPF1 (951 amino acid residue [aa]). Ci-MACPF1 consists of a signal peptide, 10 thrombospondin type 1 repeats (TSR) domains (cyan), an MACPF domain (yellow) and an epidermal growth factor (EGF)-like domain (magenta). MACPF domains are essential for cytolytic activities, as in human complement factor 6 (Hs-C6) and human perforin (Hs-Perforin). For details, see Supplementary Figure 5. **b** Domain structure of the variable-region containing chitin-binding protein, VCBP-C (349 aa). VCBP-C consists of a signal peptide, two variable-type immunoglobulin (Ig-V) domains and a chitin-binding domain (CBD) of carbohydrate-binding module family 14. Horizontal lines denote the regions corresponding to N-terminally His-tagged recombinant proteins: Wt, ΔC, and ΔV. **c** Upper panels show that VCBP-C binds to chitin beads using CBD. Recombinant proteins are visualized with chromogenic detection of His-tag (purple). Lower panels show that Wt tethers *Ciona*-gut derived bacilli on chitin beads. Recombinant proteins and bacilli were visualized with confocal laser scanning microscopy using fluorophore-conjugated anti-His tag antibody (blue) and a nuclear staining reagent, TO-PRO-3 (red), respectively. **d** Domain structure of the gel-forming mucin, Ci-GFM1 (2880 aa). Ci-GFM1 is a mosaic protein composed of 30 domains of 13 types. It shares core organization with human von Willebrand factor (Hs-VWF, 2813 aa) and human GFM (Hs-MUC2, 5179 aa). Common scaffolding consists of D1 (cyan), D2 (yellow), D`D3 (magenta), and D4 (green) assemblies, two von Willebrand C (VWC) domains and a C-terminal cystine knot (CK) domain (blue). Each protein has additional domains with specific functions. For example, CBD for chitin-binding in Ci-GFM1, von Willebrand A (VWA) domain for platelet binding in Hs-VWF and proline–threonine–serine-rich (PTS) domain for hyper glycosylation in Hs-MUC2. FN fibronectin type II domain, VWD von Willebrand D domain, C8 cysteine 8 domain, TIL trypsin inhibitor-like domain, E E domain. **e** Intermolecular disulfide bonds in D`D3 (magenta) and CK (blue) are essential for the multimeric structures, concatenating rope of Hs-VWF (left) and flat hexagonal net of Hs-MUC2 (right), which are the structural bases of the physiological functions of these molecules[18, 19]. Scale bars **c** 100 µm

share common descent (Fig. 4b). This helps us infer how gut microbes have coevolved with mucosal immune systems that maintain gut homeostasis in chordates. The co-occurrence of chitin-based barrier immunity in invertebrate outgroups and the two basal chordates, lancelets, and tunicates, indicates that an equivalent system existed in the chordate ancestor. This means that as filter-feeding non-selectively transported environmental microbes into the gut space as food (Supplementary Fig. 1 and 2, Supplementary Movies 1 and 2), these potential pathogens were confined to the luminal space and kept away from the gut epithelium, being enclosed by chitinous barrier membranes (Fig. 4a). This radial compartmentalization of the gut space, which we posit as an ancestral condition of chordates, is observed in the ray-finned fish, *O. mossambicus* (Fig. 6), but luminal microbes can be assimilated in different ways. In suspension-feeding invertebrates, including basal chordates, enzymatic digestion gradually occurs across the semi-permeable chitinous membranes, and viable passage of ingested microbes through the gut is common[37]. In contrast, the majority of ingested microbes do not reach the intestine in jawed vertebrates, including *O. mossambicus*, because of bactericidal gastric juice[38–40]. Although gut dilation for food storage occurs in invertebrates and termed stomach, as in *Ciona*, gastric secretion of hydrochloric acid is an invention of jawed vertebrates[41,42]. This gastric barrier to microbial influx appears to exert further compartmentalization of the gut space long-itudinally, thereby providing the intestine as a new ecological niche for survivors. Indeed, microbial profiling confirms dense population of non-environmental microbes in intestines of various ray-finned fishes including *O. mossambicus*[43–45] (Fig. 6c). Thus, the chitinous barrier of *O. mossambicus* likely contributes to homeostasis with indigenous microbes as the first line of defense, together with a broad representation of innate and adaptive immune responses that are largely conserved in mammals[26]. Although we consider the condition of *O. mossambicus* as a transitionary state from the chordate ancestor to mammals, this does not preclude other possible states, given the vast diversity of food-habits and anatomy in ray-finned fishes, e.g., the massive secondary loss of acidic stomach[46] (Fig. 6n, q and Supplementary Fig. 6g).

The salient feature of the mammalian gut is that chitin-based barrier immunity no longer exists, and luminal microbes directly interact with the surrounding, goblet cell-derived GFM mucus. This GFM mucus fulfills the primordial necessity of limiting microbial access to host tissue through joint actions with diffusive effector molecules (e.g., mucosal antibodies or antimicrobial peptides) in regionally diversified manners[3,47] (Fig. 6h–l). Simultaneously, this protective mucus has a role as a nutrient source for gut microbes. GFM is heavily and diversely glycosylated on its PTS domains, and these glycans are recognized and consumed by gut microbes[48]. This glycan-foraging drives microbial adaptation to this novel mucosal interface through competition for persistence[49]. Especially in the distal gut, where food-derived carbohydrate is scarce, GFM mucus forms a distinct niche for dense microbial colonization[24,50] (Fig. 6i–l). In turn, glycan-feeding enables hosts to shape microbial compositions by manipulating the glycan landscape[51]. Ecological theory predicts that this host control over microbial ecosystems was a key for establishment of the mammalian gut microbiota that is diverse, but beneficial[52]. This highlights the loss of chitin as a prerequisite for colonization of goblet cell-derived GFM mucus by indigenous gut microbes in mammals. With or without this novel type of animal–microbe association, the guts of mammals and ray-finned fish likely provide disparate mucosal environments that impose distinct selective pressures on microbial composition. This may at least partly explain why, in reciprocal transplantation of indigenous gut microbes between mice and zebrafish, transplanted

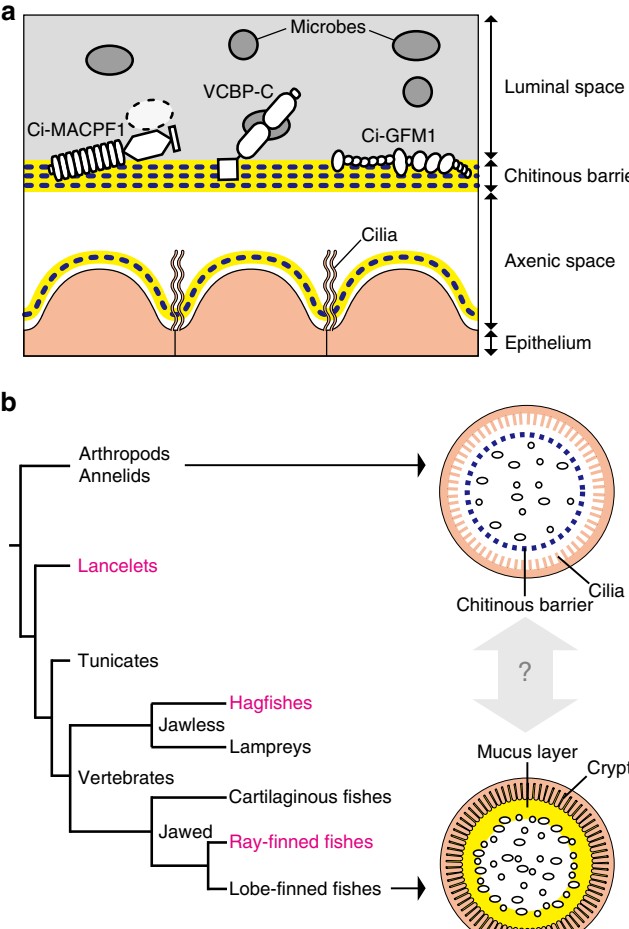

**Fig. 4** A chitin-based barrier immunity model. **a** A model. The intestinal mucosal surface of the tunicate, *C. intestinalis* Type A, is axenic due to the barrier function of multi-layered chitinous membranes that confines microbes to the luminal space. The barrier function results from sieving by a chitinous framework (blue dotted lines) and immune functions of matrix substances (yellow lines), e.g., cytolytic Ci-MACPF1 (left), bacteria-seizing VCBP-C (center), and possible multimeric protein Ci-GFM1 (right). Delamination of a new membrane from the epithelium renews axenic conditions. **b** Evolutionary implications. The tree diagram depicts phylogeny of chordates (lancelets, tunicates and vertebrates) and invertebrate outgroups (arthropods and annelids). Two arrows extending from outgroups and lobe-finned fishes point to schematic drawings of intestinal barrier immunity representative of each group. Note that these simple drawings highlight physical, but not cellular nor chemical components of barrier immunity. Invertebrate outgroups share a chitinous barrier membrane, known as the peritrophic matrix (PM, a blue dotted circle)[21,32], which encloses food matter and luminal microbes (ovals). The mammalian subgroup of lobe-finned fishes possesses a GFM-based mucus layer (a yellow circle) that covers the mucosal surface and hampers microbial access to the epithelium, while harboring dense microbes (ovals)[24,31]. The second diagram shows the distal colon of mice. The mammalian mucus system has multiple physiological roles, and the condition of mucus varies along the longitudinal axis of the intestine. Although invertebrate PMs and mammalian mucus layers are considered analogous, i.e., with no common descent, the chitin-based barrier immunity newly found in tunicates (**a**) provides molecular evidence for a possible link between these two barrier immune systems. To test this idea, animal groups that occupy intervening phylogenetic positions (typed in magenta) are critical

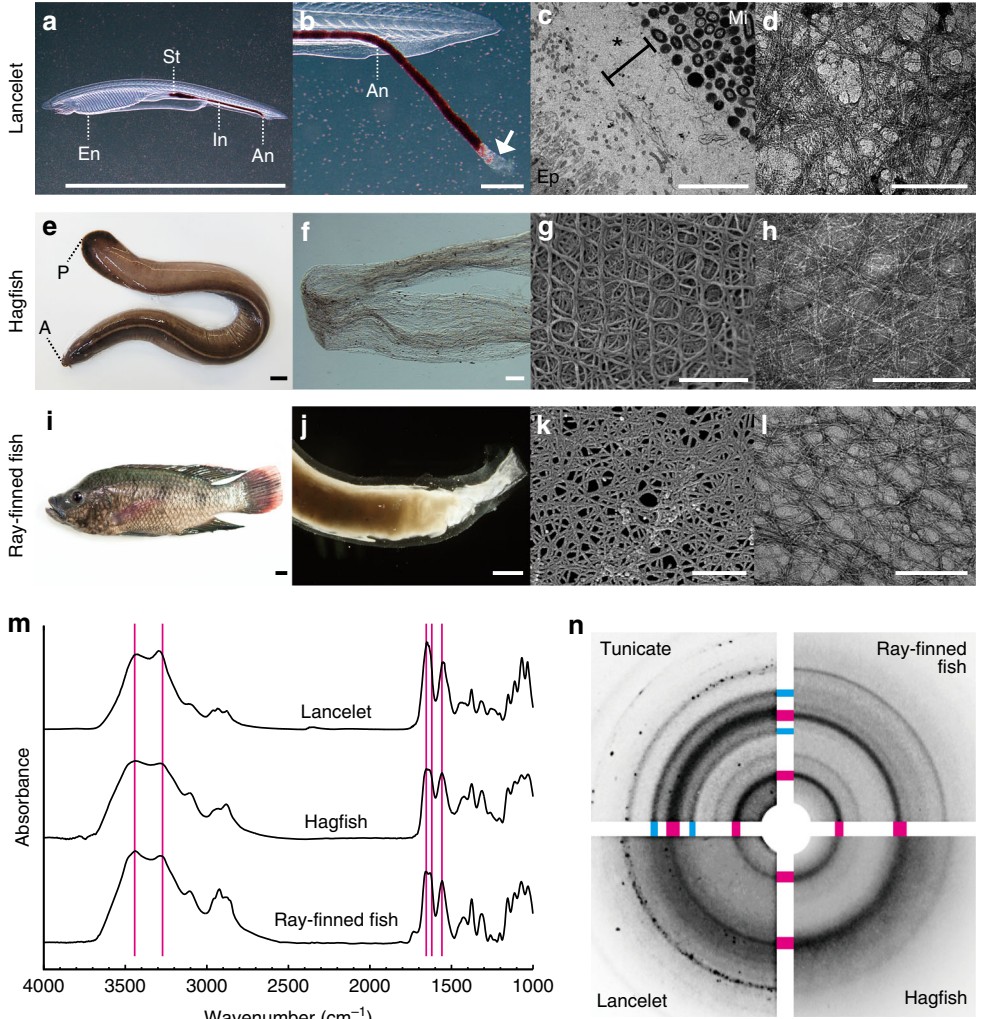

**Fig. 5** Intestinal chitinous membranes are widely distributed in chordates. **a** The lancelet, *Branchiostoma floridae*, is an internal filter-feeder that uses mucus nets, like tunicates. Mucus nets, secreted from the endostyle (En) and transported to the stomach (St) and intestine (In), are recognizable by their trapped red beads. For details, see Supplementary Fig. 2 and Supplementary Movie 2. **b** Mucus nets defecated from the anus (An) are enveloped inside a translucent membrane that forms a hollow tube (arrow). **c** A TEM image of gut cross-section showing that ingested microbes (Mi) are separated from the densely ciliated epithelium (Ep) by multi-layered membranes (*). **d** A negative-stain TEM image of a partially purified membrane showing crystalline nanofibers. **e** The hagfish, *Eptatretus atami*. The anterior and posterior ends of the body are indicated by A and P, respectively. **f** Upon starvation, *E. atami* intermittently excretes translucent empty tubes. **g** An SEM image of a purified tube showing a multi-layered, woven pattern of nanofibers. **h** A negative stain TEM image of a purified tube showing crystalline nanofibers. **i** The ray-finned fish, *Oreochromis mossambicus* (Mozambique tilapia). **j** *O. mossambicus* excretes tubular feces enveloped inside a membrane. **k** An SEM image of purified membrane showing meshed nanofibers. **l** A negative-stain TEM image of a purified tube, showing crystalline nanofibers. **m** Spectroscopic profiles of purified membranes (FT-IR spectra). Magenta vertical lines indicate peak positions specific to chitin (α allomorph). **n** Crystalline profiles of purified membranes. This composite image shows near quarter sections of the original X-ray diffractograms, combined with arcs depicting chitin-specific or cellulose-specific signals (magenta or cyan, respectively). Scale bars **a**, **e i** 1 cm; **b** 200 μm; **c** 5 μm; **d**, **g**, **h**, **k**, **l** 100 nm; and **f**, **j** 1 mm

communities change their composition to resemble that of recipients[53]. In this way, an approach to integrate microbiome data into evolutionary trajectories of host natural histories would advance our understanding of this coevolved association.

In conclusion, this comparative study provided a glimpse of gradual changes in the intestinal mucosal surface in chordates. We propose a transition from a chitin-based ancestral condition to a mucin-based derived state (Fig. 7). Concomitantly, gut microbes appear to have changed position from transient passengers that are incorporated from the surrounding environment, as food, to a selected assembly that inhabits the mucus layer as an integral part of the host fitness. We begin to appreciate that spatial organization of gut microbes lays the foundation of this microbial ecosystem[54,55]. Compartmentalization, which is usually neglected in gut homogenates prepared for microbiome studies, may give us further insight into animal–microbe associations in this digestive and the largest immune organ of the body, the gut.

## Methods
**Animals.** *C. intestinalis* Type A were supplied by the National BioResource Project [marinebio.nbrp.jp/index.jsp]. *B. floridae* and *E. atami* were collected from Tampa Bay, Florida, USA and Sagami Bay, Kanagawa, Japan, respectively. *O. mossambicus* were obtained from a stock maintained at the University of Tokyo. *D. rerio* and *O. mykiss* were purchased from local fish farmers. *C. intestinalis* Type A, *B. floridae*, and *E. atami* were maintained in separate tanks supplied with circulating filtered seawater at 18, 22, and 10 °C, respectively. *O. mossambicus*, *D. rerio*, and *O. mykiss* were maintained in separate tanks supplied with circulating filtered freshwater at 25, 20 and 20 °C, respectively. Mice were wild-type C57BL/6N (male, 10 weeks, Charles River Laboratories Japan, Inc). *C. intestinalis* Type A and *B. floridae* were

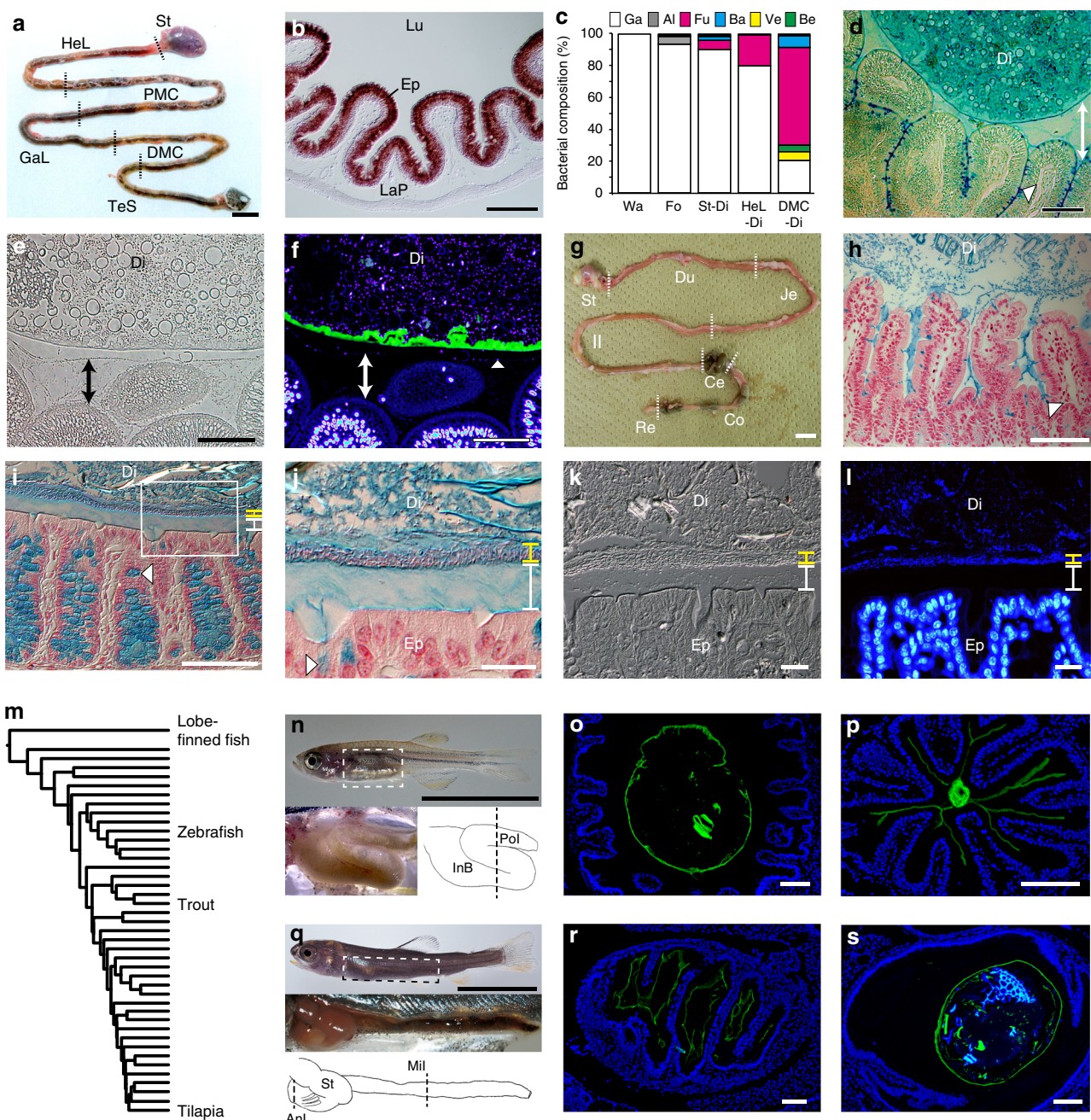

**Fig. 6** Microbial colonization is a major distinction between the mucus layers of ray-finned fish and mice. **a** Gastrointestinal tract of *O. mossambicus*. St. stomach, HeL hepatic loop, PMC proximal major coil, GaL gastric loop, DMC distal major coil, TeS terminal segment. **b** In situ hybridized cross-section of DMC, showing expression of a chitin synthase gene *Om-CHS1* in the epithelium (Ep). LaP lamina propria, Lu lumen. **c** Intestines harbor indigenous bacterial community. This panel summarizes 16S rRNA gene analysis of aquarium water (Wa), food (Fo), stomach digesta (St-Di), HeL digesta (HeL-Di), and DMC digesta (DMC-Di). Ga gammaproteobacteria, Al alphaproteobacteria, Fu fusobacteria, Ba bacteroidetes, Ve verrucomicrobia, Be betaproteobacteria. For details, see Supplementary Table 2. **d** Alcian blue-stained cross-section of DMC. Goblet cell-derived mucus fills the space between digesta (Di) and the epithelium (double-headed arrow). Digesta contains abundant mucus from pharyngeal regions. An arrowhead denotes a goblet cell. **e, f** Cross-sections of DMC (**e** Phase-contrast; **f** CBD-DAPI double-staining). A chitinous membrane (green) separates digesta microbes (blue) from the mucus layer covering the DMC epithelium (double-headed arrows). An arrowhead denotes a DAPI signal at the surface of the mucus layer. **g** Gastrointestinal tract of a mouse. Du duodenum, Je jejunum, Il ileum, Ce cecum, Co colon, Re rectum. **h–j** Alcian blue-stained gut sections counterstained with nuclear fast red (**h**, ileum; **i**, colon; **j**, the boxed area in **i**). Colon mucus covers the epithelium and consists of an inner layer devoid of microbes (white) and an outer layer densely colonized with microbes (yellow). Arrowheads show mucus granules in goblet cells. **k–l** Cross-sections of colon (**k**, DIC; **l**, DAPI). **m** Phylogeny of ray-finned fish[65], showing lobe-finned fishes (outgroup), zebrafish, rainbow trout and tilapia. **n** Zebrafish secondarily lost the stomach. The anterior intestine enlarges as the intestinal bulb (InB). PoI posterior intestine. **o, p** CBD-DAPI double-stained sections at the dotted line in **n** (**o**, InB; **p**, PoI). **q** Rainbow trout fry. The stomach (St) bends anteriorly, followed by the intestine. AnI anterior intestine, MiI middle intestine. **r, s** CBD-DAPI double-stained sections at dotted lines in **q** (**r**, AnI; **s**, MiI). For clarity, liver, gall bladder and spleen were removed in **n** and **q**. Scale bars (**a, g, n, q**) 1 cm, (**b, d–f, h, i, o, p, r, s**) 100 µm, and (**j–l**) 20 µm

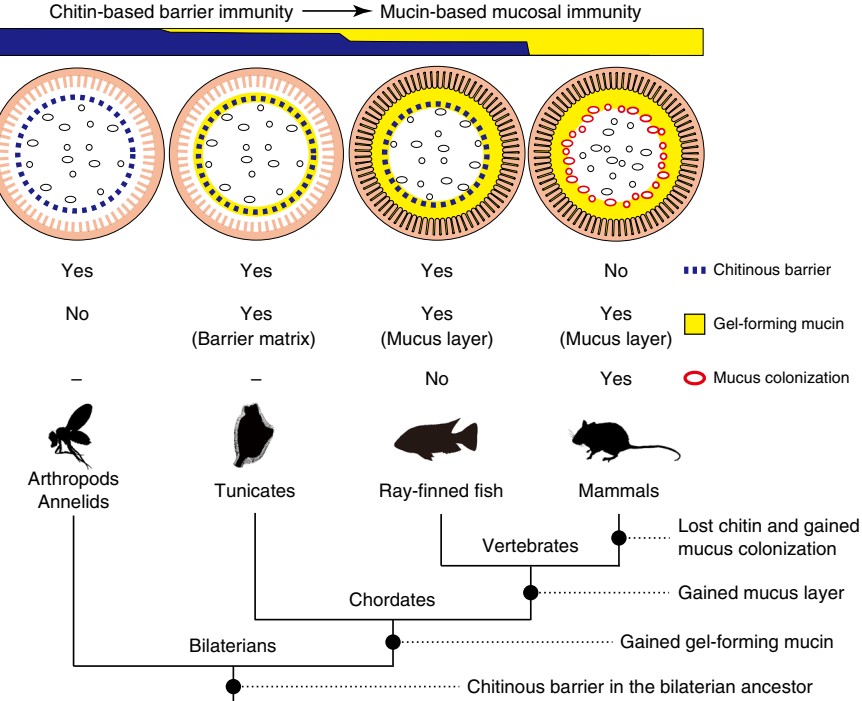

**Fig. 7** Transition of gut mucosal surface in chordates and its implication for animal–microbe association. This figure summarizes results of this comparative study of chordates. For animal groups, shown as pictograms, intestinal barrier structures are illustrated above. These illustrations focus on physical, but not cellular or humoral, components of barrier immunity. Arthropods and annelids share chitinous barrier membranes (blue dotted line) that allow movement of nutrients, but not luminal microbes (black ovals), onto the ciliated gut epithelium. This so-called peritrophic matrix (PM) is widely observed in other invertebrates, although the presence of chitin remains unclear. Tunicates possess chitinous membranes embedded in a matrix of gel-forming mucin (yellow circle). This membrane confines food microbes into the luminal space and keeps the ciliated epithelium almost axenic. In ray-finned fish, the mucosal surface is covered with a layer of gel-forming mucin that is secreted from goblet cells. This mucus layer is separated from the luminal, indigenous microbial community by chitinous barrier membranes. In mammals, chitinous membranes no longer exist, and gut microbes directly interact with the surrounding layer of gel-forming mucin. Note that the mammalian mucus system has multiple physiological roles, and there exists regional variation in mucus conditions. This illustration depicts the mouse colon, in which the mucosal surface is covered with two layers of gel-forming mucin, with the outer layer forming a distinct niche for dense microbial colonization (red ovals). "Yes" or "No" indicate the presence or absence of the items listed on the right, respectively. Previously, invertebrate PMs and mammalian mucus layers were not believed to share common descent. New data on tunicates and ray-finned fish, however, fill this gap and suggest a transition from a chitin-based ancestral condition to a mucin-based derived state (top). A tree diagram of animal phylogeny (bottom) helps to infer events that account for the transition (black circles on branches). Mucus colonization in mammalian guts appears to be a novel type of animal–microbe association that was established upon loss of chitin

fed on diatoms (*Chaetoceros gracillis*) once a day. *E. atami* can be kept without feeding for at least 3 weeks. *D. rerio* and *O. mykiss* were fed on minced fish once a day. *O. mossambicus* were fed on a commercial carp chow (Nihon Haigou Shiryo, Japan) once a day. Mice were fed a laboratory rodent diet 5001 (Japan SLC, Japan). Animal experiments were conducted in accordance with guidelines from the Okinawa Institute of Science and Technology Experimental Animal Committee.

**Isolation and purification of envelope membranes.** Adult specimens of *Ciona* were starved for 2 weeks to clear the gut contents, and then were fed with sepia ink (Liofresh, TOYO ADL, Japan) or dyed polystyrene beads (Polybead microsphere 10 μm, Polysciences, USA) for 2 h and starved again in a new tank for 4 h. Animals were anesthetized with gradual addition of 0.8% (w/v) l-menthol (Nacalai tesque, Japan) in ethanol to the seawater. Intestines were surgically isolated and transferred to a petri dish filled with phosphate-buffered saline (PBS). Intestines were longitudinally opened using scissors and forceps for microsurgery, allowing isolation of intact envelope membranes. Isolated membranes were rinsed with PBS and fixed with 4% paraformaldehyde phosphate buffer solution (PFA) (Wako Pure Chemical, Japan) at 4 °C for 1 h. Fixed membranes were washed with PBS, treated with 1 N KOH at room temperature (RT) for 18 h, washed with PBS and treated with 0.3% $NaClO_2$, buffered at pH 4.8 in acetate buffer, at 80 °C for 3 h. After several washes with ultrapure water, purified envelope membranes were kept in distilled water at 4 °C until use.

*B. floridae* were starved for 1 day to clear the gut contents. Animals were then fed with sepia ink or polystyrene beads. Feces were collected, fixed with 4% PFA at 4 °C for 1 h and washed with ultrapure water. Fixed specimens were treated with 1 N NaOH at RT for 1 h, which allowed separation of envelope membrane fragments from feces. Membrane fragments in the supernatant were collected by decantation,

followed by a centrifugation at 10,000×*g* at RT for 10 min. Precipitates were chemically purified as described above.

*E. atami* intermittently excretes a hollow tube of almost the entire gut length. Tubes were collected, fixed and chemically purified as described above.

*O. mossambicus* were fed with a chitin-free food prepared from fish meat and wheat flour for 3 days. Fish were anesthetized with 0.1% 2-phenoxyethanol and decapitated. Whole intestines were surgically isolated, fixed and chemically purified as described above.

**SEM.** Intestines were surgically isolated from anesthetized *Ciona*, cut open into small pieces (5 × 5 mm), fixed with 2.5% glutaraldehyde, 4% PFA, 150 mM NaCl, 100 mM HEPES-KOH (pH 7.2) at RT for 2 h and postfixed with 1% osmium tetroxide, 150 mM NaCl, 100 mM HEPES-KOH (pH 7.2) for 2 h on ice. Fixed specimens were dehydrated in a graded ethanol series, substituted in *t*-butyl alcohol and freeze-dried. Dried specimens were coated with osmium (Neo Osmium Coater, MeiwaFosis, Japan) and examined using a Hitachi S-4800 at an accelerating voltage of 0.5–1.0 kV. Mucus cords isolated from the dorsal lamina of *Ciona*, feces of *B. floridae* and chemically purified envelope membranes of *C. intestinalis* Type A, *B. floridae*, *E. atami*, and *O. mossambicus* were examined following this protocol.

**TEM and negative staining.** Young adult specimens (3 months) of *C. intestinalis* Type A and adult specimens of *B. floridae* were reared in natural seawater for 3 days, anesthetized with l-menthol and fixed as for SEM. Fixed specimens were dehydrated in a graded ethanol series, substituted in propylene oxide and embedded in epoxy resin, followed by polymerization at 70 °C for 16 h. Ultrathin sectioning (80 nm thickness) was done with a diamond knife (Diatome, USA) and an Ultracut UCT ultramicrotome (Leica, Germany). Sections were picked up on

Formvar-coated copper grids, stained with aqueous uranyl acetate and lead citrate and examined under a JEM 2000 EX II (Jeol, Japan) operated at 200 kV with a CCD camera (Keen view, Olympus soft image solutions, Germany). For negative staining, purified envelope membranes were mounted on carbon-coated hydrophilic grids, stained with 0.5% uranyl acetate, air-dried, and examined as described above.

**Toluidine blue staining**. Young adult specimens (3 months) of *C. intestinalis* Type A and adult specimens of *B. floridae* were fixed, embedded in epoxy resin and semithin sectioned (1 μm thickness) as for TEM. Sections were stained with 0.05% toluidine blue solution (pH 7.0) (Wako Pure Chemical, Japan) and photographed using an M205MA fluorescence microscope (Leica, Germany).

**Confirmation of axenic conditions using PCR**. Intestines were surgically isolated from anesthetized adult specimens of *C. intestinalis* Type A (*n* = 3), rinsed with PBS and transferred to a sterile petri dish. After absorbing excess PBS with sterile filter papers, the intestines were longitudinally cut open as described above, and envelope membranes were transferred to sterile petri dishes. The middle part of the anterior or posterior half of each envelope membrane were sectioned (3 mm length) and collected to separate microtubes. Pieces of the epithelium (3 × 3 mm) were excised from the corresponding part of the remaining gut tissues and collected to separate microtubes. Specimens were treated with 180 μL of 50 mM NaOH at 95 °C for 10 min, mixed with 20 μL of 1 M Tri-HCl (pH 8.0) and centrifuged at 12,500×*g* at RT for 10 min. Supernatants were collected for PCR amplification using KOD FX Neo DNA polymerase (Toyobo, Japan) and 16S universal primers: 27f (AGAGTTTGATCMTGGCTCAG) and 1492r (TACGGYTACCTTGTTACGACTT). Reaction products were assessed by agarose gel electrophoresis followed by SYBR safe staining (Thermo Fisher Scientific, USA). Amplification of 16S rRNA genes was confirmed by subcloning and Sanger sequencing of PCR products.

**FT-IR spectroscopy**. Purified membranes were deposited on a Teflon sheet and allowed to air-dry. FT-IR spectra were recorded with 4 per cm resolution and 64 scans on a Nicolet Magna 860 (Thermo Fisher Scientific, USA) in a transmission mode.

**X-ray diffraction**. X-ray diffraction patterns were obtained from purified envelope membranes using Ni-filtered Cu Kα radiation ($\lambda = 0.15418$ nm) from a rotating anode X-ray generator (RU-200BH, Rigaku, Japan) operated at voltage of 50 kV and excitation current of 100 mA. Diffraction patterns were recorded using a camera system equipped with a flat imaging plate (BAS-IP SR127, Fujifilm, Japan). The camera length was calibrated using NaF ($d = 0.23166$ nm).

**Electron diffraction**. Purified membranes were dispersed in ultrapure water by sonication (Bioruptor UCW-310, BM Equipment, Japan). Aliquots were mounted on carbon-coated hydrophilic grids and air-dried. Micro-diffraction was done with a JEM 2000 EX II operated at 200 kV. A small electron probe was generated with a condenser aperture of 20 μm and focused to a diameter of about 100 nm upon diffraction. Diffraction patterns were recorded on FDR-UR-V imaging plates (Fujifilm, Japan) with a camera length of 15 cm and an irradiation period of 1–2 s.

**MS analysis of chitinase product**. Purified envelope membranes of *C. intestinalis* Type A, *B. floridae*, *E. atami* and *O. mossambicus* were separately treated with recombinant hyperthermophilic chitinase PF-ChiA from *Pyrococcus furiosus* (Thermostable Enzyme Laboratory, Japan) following manufacturer's instructions. An aliquot (5 μL) was collected from each reaction at 0, 1, 2, 3 and 4 h after the onset of reaction and subjected to LC-MS analyses. For LC, we used an ACQUITY UPLC-I-class (Waters, USA) equipped with an ACQUITY UPLC BEH amide column and performed with a 5-min gradient from 80/20 to 25/75 MeCN/H$_2$O with 0.1% NH$_4$OH at a flow rate of 0.17 mL/min. For MS, we used a SYNAPT G2-S (Waters, USA) in ES+ ionization mode with capillary voltage at 2.0 kV, cone voltage at 25 V, source temperature at 120 °C desolvation temperatures at 350 °C, a desolvation gas flow of 500 L/h, cone gas flows of 50 L/h, scan time of 0.2 ms at a high resolution mode (35,000 full-width at half maximum) and a lock mass of Leu-enkephalin (556.27 *m/z*) at every 60 s.

**Molecular cloning of chordate chitin synthases**. Total RNA was isolated from young adult specimens (1 month) of *C. intestinalis* Type A, reverse-transcribed and used for RT-PCR[56]. Primers were designed from a transcript KH.L22.57.v1.B.ND1-1 (1481 bp) identified by tblastn searches in the ghost database[57] using invertebrate chitin synthases as queries. It was necessary to conduct rounds of rapid amplification of cDNA ends (RACE) (Marathon cDNA amplification kit, Clontech, USA) to obtain full-length transcripts (5487 bp) encoding Ci-CHS. Primer sequences used in this study are provided in Supplementary Table 3.

 *B. floridae* EST clones bfad030d07 and bflv038m04 were obtained from the cDNA database[58] by tblastn search with Ci-CHS as a query. These clones partially encoded 582 and 988 amino acid residues of putative chitin synthases and were used as queries for blastp searches at NCBI[59] to obtain hypothetical proteins

BRAFLDRAFT_68947 and BRAFLDRAFT_118918, respectively. Other putative chitin synthases of *B. floridae* were identified *in silico* elsewhere[33].

 Total RNA was isolated from whole intestines of *O. mossambicus*, reverse-transcribed and used for RT-PCR[60]. Primers were designed based on results of tblastn searches with Ci-CHS as a query in the genome assembly Orenil1.1 (http://www.ncbi.nlm.nih.gov/assembly/354508/) of the sister species, *Oreochromis niloticus*. PCR products were subcloned and sequenced as described above.

**In situ hybridization**. Young adult specimens (1 months) of *C. intestinalis* Type A were used for *in situ* hybridization with an RNA probe synthesized from full-length *Ci-CHS*, as previously reported[56].

 Juvenile specimens (~1 cm long) of *B. floridae* were used for in situ hybridization[61]. We used the EST clones bfad030d07 and bflv038m04 to synthesize probes for *BRAFLDRAFT_68947* and *BRAFLDRAFT_118918*, respectively.

 Whole intestines of *O. mossambicus* were surgically isolated and fixed with 4% PFA at 4 °C for 16 h. Fixed specimens were embedded in paraffin, sectioned (10 μm thickness) and used for in situ hybridization[60]. An RNA probe was synthesized from a DNA fragment that was amplified from *Om-CHS1* by PCR using primers, GCTCGCAGGTCAGATTAC and AGGTCTTCAGTTGTCAGGA.

**Inhibition of chitin synthesis**. Young adult specimens of *C. intestinalis* Type A (3 months, *n* = 84) were reared at 18 °C in natural seawater for 3 days. Water was switched to filtered seawater containing 30 μM Nikkomycin Z (Sigma, USA), *Serratia fonticola* (1.5x10$^4$ colony forming unit [CFU]/mL), *Staphylococcus epidermidis* (7.5 × 10$^4$ CFU/mL) and *Vibrio ezurae* (7.5 × 10$^4$ CFU/mL). Control group (*n* = 84) lacks Nikkomycin Z, and the second experiment group (*n* = 102) contains additional 100 μg/mL Streptomycin sulfate (Wako Pure Chemical, Japan). Water was changed every 3 days, and viability of animals was evaluated under light microscopy after 2 weeks.

**Proteome analysis**. An envelope membrane was isolated from an adult specimen fed with sepia ink in filtered seawater as describe above, cut open in sterile PBS and separated from a luminal mucus cord, followed by several washes with PBS. A small piece of membrane (3 × 3 mm) was excised and treated with 0.1 mL of 2.5% (w/v) lithium dodecyl sulfate/1% (v/v) dithiothreitol solution at 95 °C for 10 min. After cooling to RT, the specimen was centrifuged at 10,000×*g* for 15 min at 20 °C. Supernatant was collected, 4-fold diluted with ultrapure water and dissolved in NuPAGE LDS sample buffer (Thermo Fisher Scientific, USA). Proteins were separated on a 10% SDS-PAGE gel, size fractionated, and digested with trypsin for liquid chromatography-tandem mass spectrometry[62]. Digested peptides were analyzed using a capillary liquid chromatography system Ultima3000 (Dionex, USA) connected online to a mass spectrometer (LTQ-XL; Thermo Scientific, USA). Raw spectral data were processed using SEQUEST software to extract peak lists. Peak lists were analyzed by MASCOT against the *Ciona* protein database[63]. As control experiments, we conducted the same analyses with mucus cords isolated from the dorsal lamina. We accepted results for envelope membranes when they are concentrated more than five time than in controls. Full-length transcripts that encode the most frequently identified three components Ci-MACPF1, VCBP-C, and Ci-GFM1 were isolated by RT-PCR and RACE, as described above. Primers for RT-PCR were designed from transcripts KH.C1.45.v1.A.ND1-1, KH.C4.625.v1.A.nonSL2-1, and KH.L10838.v2.A.ND1-1, respectively.

**Binding assay of recombinant VCBP-C**. The full-length transcript of VCBP-C, which encodes 349 amino acid residues, was used as a PCR template to amplify DNA fragments encoding mature VCBP-C (Wt), a deletion mutant of C-terminal CBD (ΔC) or that of N-terminal two Ig-V domains (ΔV), which covers residues 22-349, 22–280, or 278–349, respectively. PCR products were inserted in frame between the Factor Xa recognition site and the stop codon of pCold I vector (Takara Bio, Japan) by In-Fusion cloning (Clontech, USA). We added a short linker of two glycine residues after the Factor Xa recognition site. Recombinant proteins were expressed in SHuffle Express competent *Escherichia coli* (New England BioLabs, USA), followed by cell lysis with B-Per reagent (Thermo Fisher Scientific, USA) and affinity purification with Talon resin (Clontech, USA). Purification was assessed by SDS-PAGE and western blotting using anti-His-tag mAb-HRP-DirecT antibody (Code number: D291-7, Medical & Biological Laboratories, Japan) at 1:4000 (v/v) dilution with ImmunoStar zeta chemiluminescence reagent (Wako, Japan) (Supplementary Fig. 5d). Purified recombinant proteins (4 μg each) were separately mixed with 200 μL of 10% slurry of chitin beads (New England BioLabs, USA) pre-blocked with 1% casein in PBS (Thermo Fisher Scientific, USA) and gentry rotated for 5 h at 18 °C. Beads were washed five times with PBS, mixed with the HRP-conjugated anti-His antibody in PBS (1/5000 dilution), gentry rotated for 30 min at 18 °C, washed five times with PBS and subjected to color development with TrueBlue Peroxidase substrate (KPL, USA).

 *Bacillus* sp. isolated from the gut of *C. intestinalis* Type A were grown in LB broth at 37 °C. Cells grown to mid-logarithmic phase were harvested by centrifugation at 5000×*g* for 5 min and suspended in PBS to 1.5 × 10$^5$ CFU/mL. An aliquot of cell suspension (100 μL) was mixed with equal volume of 10% slurry of casein-blocked chitin beads together with one of the recombinant proteins (4 μg). After gentle rotation for 1 h at 18 °C, beads were gently washed twice with PBS by

gravitational sedimentation and mixed with anti-His-tag mAb-Alexa Fluor 488 antibody (Medical & Biological Laboratories, Japan) in PBS (1/2500 dilution) containing 0.1% (v/v) TO-PRO-3 nuclear staining reagent (Molecular Probes, USA). After incubation for 30 min at 18 °C, beads were washed twice with PBS using gravitational sedimentation, mounted on glass slides with Vectashield mounting medium (Vector Laboratories, USA) and examined with confocal microscopy LSM 510 Meta (Zeiss, Germany).

**16S rRNA gene analysis of tilapia gut microbes**. *O. mossambicus* was fed with a chitin-free diet for a week, and the whole intestine was isolated as described above. The hepatic loop and the distal major coil were longitudinally cut open. Mucus that covers envelope membranes was removed with sterile filter papers and washed away with PBS. Envelope membranes were cut open, and luminal digesta was collected by suction and stored at −80 °C. There was no envelope membrane in the stomachs of *O. mossambicus*. Stomach digesta, aquarium water and the chitin-free food were also collected. Aliquots (20 μL) of thawed specimens were treated with NaOH, PCR-amplified, subcloned and sequenced, as for axenic PCR. Obtained sequences were used as queries for blastn searches in 16S ribosomal RNA sequence database at NCBI. We accepted results when the alignment score was over 200 and the sequence identity was above 98%.

**Alcian blue staining**. Whole intestines were isolated from *O. mossambicus* fed with chitin-free food. Coiled intestines were extended by cutting mesenteries and fixed with methacarn solution (60% methanol, 30% chloroform, 10% glacial acetic acid) for 1 h at RT, then washed with absolute ethanol and dehydrated with absolute acetone. Dehydrated specimens were embedded in methacrylate resin (*n*-butyl methacrylate:methyl methacrylate = 7:3, 1.5% of benzyl peroxide) and polymerized at 50 °C for 24 h. Semi-thin sections (1 μm thickness) were cut as for TEM, placed in a drop of water on a glass slides and heat dried at 60 °C. Glass slides were immersed in acetone to remove resin, followed by rehydration in PBS. Rehydrated specimens were treated with 3% acetic acid (pH 2.5) for 30 min at RT and stained with 1% alcian blue in 3% acetic acid (pH 2.5) (Wako Pure Chemicals, Japan) for 2 h at RT. After washing with 3% acetic acid (pH 2.5), specimens were dehydrated with an ethanol series, substituted in lemosol (Wako Pure Chemical, Japan) and mounted on glass slides with MountQuick (Daido Sangyo, Japan). Images were obtained as for toluidine blue staining.

Mice were fed with cheese that is chitin-free for 24 h. Segments of the ileum and colon were fixed with methacarn solution for 6 h at RT, and washed and dehydrated as for *O. mossambicus*. Specimens were embedded in paraffin, sectioned (5 μm thickness), de-waxed, and hydrated following standard procedures. Hydrated sections were stained and photographed as for *O. mossambicus* with an additional counterstain using 0.1% nuclear fast red solution (Muto pure chemicals, Japan).

**CBD staining**. Gut sections of *O. mossambicus* and mice were prepared as for alcian blue staining. *D. rerio* and *O. mykiss* were euthanatized by immersion in an ice bath for 10 min. After fixation with methacarn solution at RT for 1 h, specimens were embedded in paraffin and sectioned as for mice. Sections were hydrated and incubated with fluorescein-conjugated CBD (New England BioLabs, USA) in 200-fold dilution in PBS for 1 h at RT. This CBD derived from chitinase A1 of *Bacillus circulans* WL-12 and belongs to the carbohydrate-binding module family 12[64]. Sections were washed with PBS, counterstained with 4′,6-Diamidino-2-phenylindole dihydrochloride (DAPI) and mounted with SlowFade antifade reagent (Thermo Fisher Scientific, USA). Pictures were obtained using an IX71 epi-fluorescent microscope (Olympus, Japan) with an excitation filter U-MWIB3 (Olympus, Japan) for CBD or U-MWU2 filter (Olympus, Japan) for DAPI. Numbers of DAPI signal were counted in quadrat frames (50 × 50 μm, *n* = 26) after binarization using Photoshop CS6 (Adobe systems, USA).

**Expression analysis of *Om-CHS1* by RT-PCR**. Total RNA was extracted from the brain, gill, heart, liver, kidney, esophagus, stomach, hepatic loop, proximal major coil, gastric loop, distal major coil, terminal segment, rectum, muscle and skin of *O. mossambicus*, as described above. After checking RNA quality with TapeStation 4200 (Agilent Technologies, USA), RNA samples were reverse-transcribed and used for RT-PCR with the following primers: TCCTTGTGCTGGTGGTTAT and AGTTATCTCGCTGTAGTCTGAAT, as previously described[60].

**Data availability**. DNA sequences of *Ci-CHS*, *Ci-MACPF1*, *Ci-GFM1*, and *Om-CHS1* were deposited in DNA Data Bank of Japan with accession codes LC072663, LC072664, LC072665, and LC072666, respectively. DNA sequences of 16S rRNA gene amplicons were deposited with accession codes LC409535-LC410039 and LC410204-LC411764. Proteome data set, including raw data files, processed peak lists, and database search results, are deposited at jPOST with the ProteomeXchange code, PXD010503. The data that support the finding of this study are available from the corresponding author upon reasonable request.

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

## Acknowledgements
We thank Masahisa Wada for technical assistance in X-ray diffraction, the National Bio-Resource Project for providing *C. intestinalis* Type A and Steven. D. Aird for editing the manuscript. This work was supported by the R&D Cluster Research Program of Okinawa Institute of Science and Technology Graduate University.

## Author contributions
K.N. conceived the research. S.K., Y.O., M.W., A.V.-B. and K.N. conducted experiments and analyzed data on chitin. S.K., N.S., and K.N. conducted experiments and analyzed data on *Ciona*, zebrafish and rainbow trout. S.W., S.S., T.K., S.K. and K.N. conducted experiments and analyzed data on *Mozambique tilapia*. C.-H.T., T.-M.L., J.-K.Y. and K.N. conducted experiments and analyzed data on *Branchiostoma*. L.Y., H.S. and K.N. conducted experiments and analyzed data of proteomics. K.N. wrote the manuscript.

## Additional information

**Competing interests:** The authors declare no competing interests.

