## [Peer Review file · Nature Communications]

Reviewers' comments:

Reviewer #1 (Remarks to the Author):

Below a review of the manuscript by Dr Nakashima and co-workers entitled "Hidden Chitin Illuminates the Origin of Mammalian Gut Microbiota"

The manuscript describes some hypotheses that are of interest for scientists in the field of evolutionary research of development. However, my general feeling is that the authors do not show sufficient evidence for many of the conclusions as detailed below. Therefore the manuscript remains very speculative.

1. Evidence for *Ciona intestinalis* polysaccharides being identical to chitin is not conclusive. (a) Is the CBD really specific? Control for binding to other beta glycans is needed. (b) is the Ci-Chs really producing chitin, this is speculative since homologs of Chs also can produce other beta glycans, like in vertebrate Has genes. (c) Chitinase preparations are usually not very pure and contain other hydrolases. To solve these problems and give additional evidence for the chemical nature of the polymer the best would be to add mass spectrometry data for degradation products.

2. it is not proven that the presumed chitin provides a barrier based on the Nikkomycin Z data. It is well possible that Nikkomycin Z has toxic effects or secondary effects that explain the effect on survival. A strong line of evidence would be to grow in the presence of Nikkomycin Z under aseptic conditions and shown that in this case there is no effect on survival. Or show that infection really is enhanced in the tissue leading to detrimental effects like local necrosis or damage in infected zones. As it stands Fig. 1J is very non-informative and quantitative analysis is essential.

3. The method for showing evidence for axenic conditions in *Ciona* epithelia by PCR (Suppl. Fig.1) have to be strengthened. At least providing evidence that the harsh conditions of treatment (50 mM NaOH, 95 degrees Celsius) are not destructive to the 16S RNA (I.e. Bioanalyzer - provide a RIN number to show template integrity).

4. For none of the three proteins identified with proteomics there is strong evidence for antimicrobial activity. The evidence is based on supp. Fig.3e, which shows some marginal activity. But many controls are missing. Concentrations, dose dependency, control for inactivated protein. Also other bacterial species should be tested, preferably representatives of bacteria that have been shown to be present in *Ciona*. In conclusion this part with such limited data is so speculative that it could be removed from the paper without blocking the model.

5. In the fish: same problem of lack of conclusive evidence as above with the statement of the presence of chitin as under point 1. Although there are a few publications on the presence of chitin in vertebrates, the authors don't show sufficient additional evidence to give further understanding how this presumed chitin polymer is produced. 6. The statement on absence of mucus colonisation in fish, should be made quantitative.

7. Homology of the *Ciona* and fish presumed chitin synthase genes in the study, with the published zebrafish chitin synthase (Joyce et al. 2015), should be shown and discussed. In mammals there has been evidence as to the function of hyaluronate synthase (HAS) genes in the synthesis of chitin and therefore these enzymes should also be included in the comparisons.

8. The methods used for chitin detection should be also tested in a mammalian gut systems such a rodents. Otherwise the claim that chitin is not present in mammals is just speculation based on the lack of the published opposite result.

9. Supplemental figure 8:

qPCR data does not provide any data on RNA integrity (Bioanalyzer/RIN number is essential) and does not evaluate the house keeping gene before using it (assessment by GEnorm would be standard (see DOI: 10.1186/gb-2002-3-7-research0034 for a useful reference)).

10. On P6 L22-24 P, plus p23-24 (methods): Rgd. VCBP-C.

"It has been suggested that the C-terminal chitin-binding domain (CBD) recognizes chitinous parts of non-self organisms. By contrast, our data indicate that CBD recognizes endogenous chitin in the gut membrane."

This is a problematic statement to make if you don't make sure you've gotten rid of microbial

carry-over in the proteomic analysis.

Reviewer #2 (Remarks to the Author):

The core assumptions made in this study are 1) that there are relatively conserved macro-level protective structures that are present in the gastrointestinal tracts of differentiated organisms, and 2) that 'higher' organisms have lost a more primitive means of protecting the mucosal epithelial surface from bacteria, namely a chitinous membrane with pores that allow movement of nutrients, but not pathogens (fungi, bacteria and viruses) onto the epithelial surface. It makes some assumptions about the conservation of structures that the literature don't necessarily bear out. While indeed the chitin-based membrane may be a feature of 'lower' forms, it is my reading that the "barrier" function ascribed to mucous is as much about nutrition, as it is innate immunity. For example, studies in knockout mice suggest that Muc-1, a key gastric mucin, are more sensitive to colonisation by some bacteria (Gastroenterology 133:1210), but not others (Helicobacter 13:1523), both of which are inflammatory and pathogenic. We know too that, in the gastric mucosae, the acid runs through "channels" in the mucous layer of 5-7µm (Gastroenterology 118:1297), suggesting a complexity to mucous that goes beyond the cartoons drawn in the paper (e.g. Fig 4). In short, mucous may have multiple roles and it may reduce the number of bacteria attaching to cell surfaces when coupled with peristalsis, but it also has physiological roles that go beyond a simple barrier function.

While mucous overlies the gastrointestinal epithelium, it is not necessarily a complete barrier to microbes - the degree to which this is true may be different in different regions of the gut, and vary for the types of bacteria studied. Proteobacteria through flagellae (Ann Rev Micro 2011:389), or bacteria using spirochaetal morphology (e.g. Biophys J 2006:3019), can enter and even transverse the viscous fluids such as those in the mucous layer, as can segmented, filamentous bacteria (SFBs). Scanning EMs of the gut show a variety of interactions where the bacteria sit in the mucous or between the mucous and the surface of epithelium, indeed sometimes bound to the epithelium (e.g. Gut 56:343) and SFBs intimately associate with the epithelium (Nature 520:99). The degree of 'impenetrability' of the mucous layer varies between mammals (Gut 63:281) suggesting that the rules regarding mucous protection of the epithelium are not fully evolved. Indeed, there be development of different mammalian systems, some involving two mucous layers (PNAS 105:15064).

It is not safe to assume that a mesh with pores up to 80-90nm will filter most marine viruses; indeed many viruses are less than 50nm in diameter, especially viruses which are evolved to live in harsher environments and are non-enveloped. The chitinous meshes would be capable of retaining most bacteria of course, assuming they are completely patent.

Histology is used to demonstrate retention of the macro gut contents. The arrowed spaces between membranes (F1c, SF1g) might be sensitive to fixation artefacts - have these spaces been seen in more conservative histological methods and are they truly bacteria-free?

Supplementary figure 1 is a key figure and should be in the main body of the paper.

The study speculates on the purpose of the 'chitinous bags'. While they may have a limited role in protecting the epithelium from e.g. viruses, the presence of anti-bacterial and potentially aggregating proteins within the enclosure may facilitate digestion of the bacteria that enter the organism - their function could be nutritional rather than protective.

The presence of the pore forming protein (MACPF1) could imply a nutrition-based purpose, rather than an innate defence purpose. The bactericidal assay presented is unconvincing (SF3e) - the protein has no action against E. coli, there is no dose ranging, and evidence that the protein per se is responsible is not presented. Does the pore forming protein require activation, for example,

reduction...? These core issues are not addressed. An antibacterial protein would need to show more efficacy against recognised pathogens of the organisms under investigation for the data to be convincing. Equally, are the bacteria killed or simply in stasis; it is very difficult to tell from the simple data presented. The choice of bacteria, an aerobe which forms spores, is also curious.

Apart from these concerns, the paper looks sound and the biochemical/physical analyses of the various structures is convincing. The paper is by its nature descriptive and somewhat speculative and would be improved by mutagenesis studies which removed one of more actors from the story.

Minor issues - understanding the microbiota is not simply a question of numbers. Different bacteria interact very differently with the host and the major taxonomic units provide little insight into whether the bacteria have similar or different mucosal interactions - e.g. a flagellated vs non-flagellated bacterium. It is wrong to consider the microbiota of plants or animals as a fixed community. The evolution of the microbiota follows a succession of species, each more fit the incumbent species, reprogrammed by exposure to compounds which grossly affect the organism (e.g. antibiotics). It would be interesting to see what happens when mutations in the various genes encoding proteins active in the chitinous barrier are made.

Comments from Reviewer #1: The manuscript describes some hypotheses that are of interest for scientists in the field of evolutionary research of development. However, my general feeling is that the authors do not show sufficient evidence for many of the conclusions as detailed below. Therefore the manuscript remains very speculative.

Response: We thank this reviewer for reviewing the manuscript carefully and for offering insightful comments. We believe that our additional experimental data included in the revised manuscript will address the raised concerns.

*1. Evidence for *Ciona intestinalis* polysaccharides being identical to chitin is not conclusive. (a) Is the CBD really specific? Control for binding to other beta glycans is needed. (b) is the Ci-Chs really producing chitin, this is speculative since homologs of Chs also can produce other beta glycans, like in vertebrate Has genes. (c) Chitinase preparations are usually not very pure and contain other hydrolases. To solve these problems and give additional evidence for the chemical nature of the polymer the best would be to add mass spectrometry data for degradation products.*

Response: Following this suggestion, we added mass spectrometry data for degradation products (Fig. 2f, Supplementary Fig. 3). We successfully detected release of N-acetylglucosamine (NAG) and N-acetylchitobiose (di-NAG), which are the degradation products expected when chitin is hydrolyzed by chitinase. A mass spectrometry expert who performed this experiment is now a co-author (A. V. Briones). We appreciate this technical suggestion.

Although we believe that this data directly respond to the reviewer's request, mass spectrometry data for degradation products serves as additional, though indirect support for chitin (any presence of chitin without content profiles). What provides "*evidence for *Ciona intestinalis* polysaccharide being identical to chitin*" is the combination of chemical and physical analyses: Fourier transform infrared spectroscopy (composition), X-ray diffraction (crystallographic information), SEM (nanofibrous morphology plausible for natural chitin) and TEM and electron diffraction (local crystallographic information from a single fiber) (Fig. 11, Fig. 2a–f). This point should not be overlooked. Although this comment seemed to imply that CBD was used to prove chitin in our original manuscript, this is a misunderstanding. Actually, we used CBD to provide additional visual information (bulk morphology) about the chemically purified framework of the *Ciona* intestinal membrane, which had already been proven to be chitin by chemical and physical analyses. To avoid possible misunderstanding, we added an explanation about what is learned from each analysis (P5, lines 1–18).

Additionally, misunderstanding in Minor comments a-c seems to underlie the feeling that “*the manuscript remains very speculative.*”

(a) Is the CBD really specific? Control for binding to other beta glycans is needed.

Response: No, CBD is not specific to chitin. CBD binds not only to other beta glycans, but also to glycosaminoglycan and N-glycans on glycoproteins (e.g. Ujita *et al.*, 2003 *Biosci. Biotechnol. Biochem.* **67**, 2402-2407). Consequently, even with control for beta glycans, CBD cannot be used to verify the presence of chitin in chordate specimens full of glycosaminoglycans and glycoproteins. This is why we did not use CBD, but conducted the aforementioned chemical and physical analyses to demonstrate chitin in *Ciona* and in other chordate specimens. We will further extend the discussion of CBD in our response to Comment 5.

(b) is the Ci-Chs really producing chitin, this is speculative since homologs of Chs also can produce other beta glycans, like in vertebrate Has genes.

Response: To the best of our knowledge, there is no evidence for the statement that “*homologs of Chs also can produce other beta glycans, like in vertebrate Has genes,*” if the reviewer means that vertebrate HAS (hyaluronan synthase) can produce hyaluronan and also other beta glycan(s), e.g. chitin. The plausible background for this comment may be an early controversy in 1990’s regarding functions of vertebrate HAS that was originally called DG42 (reviewed in Varki, *PNAS* **93**: 4523-4525, 1996). Some groups reported that DG42 produces hyaluronan, a glycopolymer consisting of alternating units of β 1-4-linked N-acetylglucosamine (NAG) and β 1-3-linked glucuronic acid (GlcUA). On the other hand, another group claimed that DG42 can produce chitin, which is a β 1-4-linked homopolymer of NAG, based on their finding that crude embryo extracts produced chito-oligomers.

This controversy was resolved in subsequent reports using recombinant HAS proteins of high purity. Recombinant mouse HAS, in the presence of UDP-NAG and UDP-GlcUA, produced abundant hyaluronan, but no chito-oligomers nor chitin (Yoshida *et al.*, *J. Biol. Chem.* **275**, 497-506, 2000). This is also true of recombinant HAS of bacterial origin (Weigel *et al.*, *Glycobiology* **25**: 632-643, 2015). Only when the reaction contains UDP-NAG, but lacks UDP-GlcUA, the mouse and bacterial HASs produce chito-oligomers, the degree of polymerization (DP) of which are up to 14 or 15, respectively. Because these values are far smaller than the DP of chitin, 6,400-15,700 (Kumar, *React.*

Funct. Polym. **46**: 1-27, 2000), the production of short oligomers is considered an aberrant function of HAS in the absence of one of its two natural substrates.

Another background leading to the controversy was that DG42 showed sequence similarity to both HAS and CHS (chitin synthase). Thanks to molecular phylogenetic studies using massive genome information (e.g. Zakrzewski *et al*, *Genome Biol. Evol.* **6**: 316-325, 2014), it is now clear that 1) DG42 is a member of HAS, 2) HAS and CHS are members of the glycosyl transferase family 2 (GT2), which includes various glycopolymer synthases, 3) HAS and CHS form their own clades within the molecular phylogeny of GT2, and 4) HAS-clade members produce hyaluronan but no chitin, while CHS-clade members produce chitin, but no hyaluronan. Additionally, chordate CHSs, including Ci-CHS, form a sub-clade within a clade of animal CHS in our analysis (Supplementary Fig. 4).

In summary, there is no evidence to suggest that the chordate CHS produces any beta glycan other than chitin. This could be tested in the future using recombinant CHS of high purity, as in the above case for HAS. We hope that our explanation will dispel misunderstanding of the reviewer's view on CHS.

(c) Chitinase preparations are usually not very pure and contain other hydrolases.

Response: We had the same concern, so we tested several chitinase products for purity. The one selected for our experiment is a recombinant hyperthermophilic chitinase from the Archean, *Pyrococcus furiosus* (Oku and Ishikawa, *Biosci. Biotechnol. Biochem.* **70**: 1696-1701, 2006). Combination of heat-resistance, tag-based affinity purification, anion exchange chromatography, ammonium sulfate precipitation, and gel filtration yielded a preparation presenting a single band of the expected size for this chitinase (53 kDa), as revealed by SDS-PAGE. Nevertheless, the possible presence of other hydrolases remains, although it seems rather unlikely. It is therefore advisable not to rely on chitinase in order to demonstrate chitin. It is the combination of chemical and physical analyses that confirm it, as discussed in our response to Comment 1.

2. it is not proven that the presumed chitin provides a barrier based on the Nikkomycin Z data. It is well possible that Nikkomycin Z has toxic effects or secondary effects that explain the effect on survival. A strong line of evidence would be to grow in the presence of Nikkomycin Z under aseptic conditions and shown that in this case there is no effect on survival. Or show that infection really is enhanced in the tissue leading to detrimental effects like local necrosis or damage in infected zones. As it stands Fig. 1J is very non-informative and quantitative analysis is essential.

Response: Following this suggestion, we conducted Nikkomycin Z treatment (30 μ M) on *Ciona* under aseptic condition using Streptomycin (100 μ g/ml). The survival rate was 83.3% (N=102), which is comparable to 76.2% (N=84), the rate of a group reared without both chemicals. Comparatively, the survival rate under Nikkomycin Z treatment without Streptomycin dropped to 4.8% (N=84). These results show that Nikkomycin Z has almost no toxic effect on the survival of *Ciona*, similar to the case of amphibians (Holden *et al.*, *Fungal Biol.* **118**: 48-60, 2014). These results are shown in Fig. 2j and in the Results section (P6, lines 1–9). We appreciate this technical suggestion for additional evidence of a barrier function of endogenous chitin.

3. The method for showing evidence for axenic conditions in Ciona epithelia by PCR (Suppl. Fig.1) have to be strengthened. At least providing evidence that the harsh conditions of treatment (50 mM NaOH, 95 degrees Celsius) are not destructive to the 16S RNA (I.e. Bioanalyzer - provide a RIN number to show template integrity).

Response: There is a simple misunderstanding. What was amplified in our PCR was not 16S rRNA, but DNA encoding it. We agree with the concern over RNA degradation, because the covalent backbone of RNA is rapidly hydrolyzed under alkaline conditions. The 2' hydroxyl group in RNA acts as a nucleophile in an intramolecular displacement. The 2'3'-cyclic monophosphate derivative is further hydrolyzed to give a mixture of 2'- and 3'-monophosphate derivatives. Conversely, DNA, which lacks 2' hydroxyls, is stable under similar conditions. This is the reason that alkali and heat treatments similar to ours are widely used in preparation of DNA templates for PCR in 16S rRNA “gene” analysis.

4. For none of the three proteins identified with proteomics there is strong evidence for antimicrobial activity. The evidence is based on supp. Fig.3e, which shows some marginal activity. But many controls are missing. Concentrations, dose dependency, control for inactivated protein. Also other bacterial species should be tested, preferably representatives of bacteria that have been shown to be present in Ciona. In conclusion this part with such limited data is so speculative that it could be removed from the paper without blocking the model.

Response: Following this suggestion, we removed the corresponding data from the revised manuscript (Supplementary Fig. 3d-e in the original manuscript). We appreciate this advice.

5. In the fish: same problem of lack of conclusive evidence as above with the statement of the presence of chitin as under point 1. Although there are a few publications on the presence of chitin in vertebrates, the authors don't show sufficient additional evidence to give further understanding how this presumed chitin polymer is produced.

Response: Because this comment raised the same concern as in Comment 1, which argued the best solution would be to add mass spectrometry data for degradation products, we added mass spectrometry data for degradation products of the intestinal membrane of tilapia, lancelet and hagfish. All membranes released NAG and di-NAG upon chitinase treatment (Supplementary Fig. 3). Again, we emphasize that it is the combination of chemical and physical analyses that suffices demonstration of chitin (Fig. 5), as explained in our response to Comment 1.

Regarding the second point, it seems important to summarize what was known about vertebrate chitin and what is new in our study, because there seems to be misunderstanding. We first explain that erroneous use of CBD and its precedents repeatedly caused confusion in chitin research. We believe this viewpoint is imperative for sound development of this field. We then discuss how the present work should advance the field.

Although chitin has been long believed to be absent in chordates, there were a few reports on the presence of chitin in both invertebrate- and vertebrate-chordates (Peters, *Z. Morph. Tiere* **62**, 9-57, 1968; Sannasi & Hermann, *Experientia* **26**, 351-352, 1970; Wagner *et al.*, *Experientia* **49**, 317-319, 1993; Dishaw *et al.*, *Nat. Commun.* **7**, 10617, 2016). These reports, however, relied upon staining methods using iodine (in the so-called chitosan test), wheat germ agglutinin (a lectin that binds to NAG and sialic acid), calcofluor-white (a fluorescent dye that binds to several beta-1,3 and beta-1,4 glucans) or CBD, all of which are positive for, but non-specific to chitin. It would be technically irrelevant to use these molecules, which lack specificity for chitin, to demonstrate the presence of chitin. Actually, when the same species in these reports were later reexamined by rigorous analytical methods, there were always negative results (*e.g.* Rudall and Kenchington, *Biol. Rev.* **48**, 597-633, 1973; Rähr, *Zoomorphology* **99**, 167-180, 1982). Still, these molecules continue to be viewed by some as being chitin-specific, *e.g.* calcofluor-white as “a chitin-specific general histological stain (Dishaw *et al.*, 2016).” A corollary of this misidentification of specificity would be misinterpretation of staining data, which undoubtedly leads to erroneous conclusions. In these staining methods, the presence of chitin is sufficient to raise staining signals; however, staining signals does not necessarily mean that there is chitin. To terminate this recurring confusion, we explained the technical limitations and proper use of these molecules (P10, lines 7–18). At the same time, we emphasized that these molecules can

provide practical means to assess “possible” chitin *in situ*, while *in situ* information is usually lost in chemical purification processes prior to analytical proof of chitin (P9, lines 18–22; P10 lines 17–18). We hope that researchers, especially newcomers to this emerging field, can avoid the erroneous use of staining methods, but take advantages of both the staining and the analytical methods.

There is only one case in which chordate chitin was verified based on analytical data (Tang *et al.*, *Curr. Biol.* **25**: 897-900, 2015). The authors collected a liter of scales by scraping the bodies of Atlantic salmon raised in a farm, and purified 1.5 mg of precipitate that gave a spectrum for chitin under microscopic FTIR. They also suggested the presence of chitin in the gut of zebrafish and regenerating skins of axolotls (salamander), based on expression of *CHS* and CBD-staining, but without analytical data. We agree that chitin is endogenously produced in vertebrates. However, strictly speaking, or according to the criterion of Reviewer Comment 1, an FTIR spectrum alone is “*not conclusive*.” Indeed, there remain open questions: an explanation for the morphology of precipitates consisting of distinct granules embedded in a film, which appears unusual for chitin; analytical data for deacetylation status of the precipitates (not CBD, which would represent a circular argument); how to avoid contamination with minute, epizoic parasites that are chitinous and abundant in fish farms; where the polystyrene contamination come from? In our opinion, these questions will be resolved with proper analytical data. We therefore carefully organized analyses to confirm the presence of chitin: SEM to confirm fine morphology of nanofibers that is expected to native chitin, FTIR to confirm chemical composition, X-ray diffraction to examine crystalline profiles, TEM negative staining to ascertain crystalline states of nanofibers, electron diffraction to prove crystalline profiles at a single-fiber resolution, and mass spectrometry to assess degradation products by chitinase (Fig. 11, Fig. 2a-f, Fig. 5, and Supplementary Fig. 3). Meeting several independent criteria, our analytical data verified the existence of chitin in both invertebrate- and vertebrate-chordates with an unprecedented level of reliability.

Once chitin was discovered in chordate guts, the next step to “*understand how this chitin polymer is produced*” would be to check if CHS works there, because as discussed in Minor comment 1b, there is so far no exception to the rule that chitin is produced by CHS. We thus isolated *CHS* of *Ciona*, examined its expression pattern by *in situ* hybridization, and analyzed its function using a CHS-specific inhibitor (Supplementary Fig. 4, Fig. 2h–j). One approach to further understand “*how this chitin polymer is produced*” could be structural analysis of Ci-CHS, as crystal structures of cellulose synthase provided new insights into molecular mechanisms of cellulose synthesis (Morgan *et al.*, *Nature* **531**: 329-334, 2016). This is simply out of the scope of the present study. Instead, aiming to conduct comparative analyses as widely as possible, we expanded our analysis to other chordate

lineages that occupy phylogenetically critical positions, *i.e.* lancelet, hagfish, ray-finned fish and mouse (Fig. 5, Fig. 6, Supplementary Fig. 6). As results, we were able to provide “*sufficient additional evidence to give further understanding*” of the role of chitin at the intestinal animal-microbe interface during chordate evolution.

6. The statement on absence of mucus colonisation in fish, should be made quantitative.

Response: Following this suggestion, we quantified mucus colonization in fish by counting DAPI signals in quadrat frames (50 μm \times 50 μm each) on gut sections. The average counts for the luminal space enclosed by the chitinous membrane was 95.9 ± 29.0 (N=26). Conversely, the counts for the space between the chitinous membrane and the gut epithelium was 0.6 ± 1.0 (N=26). Accordingly, it can be safely said that fish gut microbes are mostly confined to the luminal space by the chitinous membrane. We mentioned this data in the Results section (P9, lines 4–11). We appreciate this technical suggestion for quantitativity.

7. Homology of the Ciona and fish presumed chitin synthase genes in the study, with the published zebrafish chitin synthase (Joyce et al. 2015), should be shown and discussed. In mammals there has been evidence as to the function of hyaluronate synthase (HAS) genes in the synthesis of chitin and therefore these enzymes should also be included in the comparisons.

Response: Following this suggestion, we first conducted domain structure and amino acid conservation analyses and confirmed that the isolated chordate *CHSs* encode molecular features of CHS (Supplementary Fig. 4a-b and Supplementary Fig. 6a). We then conducted molecular phylogenetic analysis of animal CHS. As results, the isolated chordate CHSs and the published zebrafish CHSs were grouped together, following the pattern of organismal phylogeny (Supplementary Fig. 4c). This data forms a basis of our discussion of invertebrate chitinous membranes (P8, lines 3–4). Additionally, for readers with further interest in this topic, we cited the work by Zakrzewski *et al.* (reference 33), which conducted rigorous molecular phylogenetics of animal CHS including both the published and unpublished CHSs of zebrafish. Regarding the latter part of this comment on HAS, it may stem from misunderstanding, as we already noted in Minor comment 1b.

8. The methods used for chitin detection should be also tested in a mammalian gut systems such a rodents. Otherwise the claim that chitin is not present in mammals is just speculation based on the lack of the published opposite result.

Response: Following this suggestion, we expanded our analysis to mice. First, we conducted chemical purification, but we were not able to find any trace of chitin in the gastrointestinal tube of mice. This contrasts with the results of the same treatment on tunicates, lancelets, hagfish and tilapia, in which we obtained chitinous membranes (Fig. 1l, Fig. 2c, Fig. 5d,f–h,k,l). Second, we examined the presence and/or distribution of possible chitin, mucus and gut microbiota by staining gut sections using CBD, Alcian blue and DAPI, respectively. Note that we used CBD for obtaining *in situ* information of possible chitin, but not for demonstration of chitin, as we already discussed in the Comment 5. Results are: 1) we did not observe possible chitinous membranes that overlies the mucus layer, 2) the colonic epithelium is covered with the mucus layer secreted from crypt goblet cells and 3) while the inner part of this layer rarely contains microbes, the outer part is densely colonized with microbial communities (Fig. 6g-l). Last, despite our effort to find a mammalian CHS gene in public databases, we found no candidate gene. Thus, it can be safely concluded that mice have no chitinous membrane. We appreciate this technical suggestion that reinforced our view on mammals.

9. Supplemental figure 8: qPCR data does not provide any data on RNA integrity (Bioanalyzer/RIN number is essential) and does not evaluate the house keeping gene before using it (assessment by GEnorm would be standard (see DOI: 10.1186/gb-2002-3-7-research0034 for a useful reference)).

Response: Following this suggestion, we evaluated the integrity of our RNA samples using TapeStation/RIN^e number, which is the successor of the Bioanalyzer/RIN number. The resultant mean RIN^e number, 9.0±0.43 (N=15), showed high quality of our samples. We further omitted qPCR and concentrated on RT-PCR, because it is RT-PCR that is essential to our discussion. As a positive control, we used *18S rRNA*, which is the most stably expressed gene across various tissues in tilapia, as evaluated using geNorm (Yang *et al.*, *Gene* **527**: 183-192, 2013). Results are shown in Supplementary Fig. 6q. We appreciate this technical suggestion to improve the quality of RNA experiments.

10. On P6 L22-24 P, plus p23-24 (methods): Rgd. VCBP-C.

"It has been suggested that the C-terminal chitin-binding domain (CBD) recognizes chitinous parts of non-self organisms. By contrast, our data indicate that CBD recognizes endogenous chitin in the gut membrane."

This is a problematic statement to make if you don't make sure you've gotten rid of microbial carry-over in the proteomic analysis.

Response: We originally pointed out that there are different views on the function of CBD between two groups. We now know that two groups have the same opinion based on new data (Dishaw *et al.*, 2016). Accordingly, we removed the corresponding statement from the revision. Regarding the last part of this comment, we had the same concern and thus prepared our proteomic sample with care. First, we used *Ciona* specimens fed with sepia ink in sterile-filtered seawater (P20, lines 1–3). Second, it is technically feasible to excise a small piece (3×3 mm) of chitinous membrane while avoiding the mucus cord that confines trapped matter. As results, when we assessed microbial contamination of excised pieces by DAPI staining, we obtained no bacterial signal. We appreciate this technical comment.

Reviewer #2 (Remarks to the Author):

1. The core assumptions made in this study are 1) that there are relatively conserved macro-level protective structures that are present in the gastrointestinal tracts of differentiate organisms, and 2) that 'higher' organisms have lost a more primitive means of protecting the mucosal epithelial surface from bacteria, namely a chitinous membrane with pores that allow movement of nutrients, but not pathogens (fungi, bacteria and viruses) onto the epithelial surface. It makes some assumptions about the conservation of structures that the literature don't necessarily bear out. While indeed the chitin-based membrane may be a feature of 'lower' forms, it is my reading that the "barrier" function ascribed to mucous is as much about nutrition, as it is innate immunity. For example, studies in knockout mice suggest that Muc-1, a key gastric mucin, are more sensitive to colonisation by some bacteria (Gastroenterology 133:1210), but not others (Helicobacter 13:1523), both of which are inflammatory and pathogenic. We know too that, in the gastric mucosae, the acid runs through "channels" in the mucous layer of 5-7um (Gastroenterology 118:1297), suggesting a complexity to mucous that goes beyond the cartoons drawn in the paper (e.g. Fig 4). In short, mucous may have multiple roles and it may reduce the number of bacteria attaching to cell surfaces when coupled with peristalsis, but it also has physiological roles that go beyond a simple barrier function.

Response: We thank this reviewer for reviewing the manuscript carefully and for offering insightful comments. We agree that the mammalian mucus system has multiple physiological roles above and beyond a simple barrier function. Accordingly, we changed statements in corresponding parts of the revised manuscript so as to acknowledge this fact. For instance, where the first cartoon of the mammalian mucus system appears (Fig. 4b), we stated that “Note that these simple drawings highlight physical, but not cellular nor chemical components of barrier immunity (P34, lines 7–8)...The mammalian mucus system has multiple physiological roles, and the condition of mucus varies along

the longitudinal axis of the intestine (P34, lines 13–15)”. Similarly, in the summary figure (Fig. 7), we stated that “Note that the mammalian mucus system has multiple physiological roles, and there exists regional variation in mucus conditions. This illustration depicts the mouse colon, in which the mucosal surface is covered with two layers of gel-forming mucin, with the outer layer forming a distinct niche for dense microbial colonization (P37, lines 13–17).” We believe that these new statements indicate that we are discussing but one of the various functions of a structurally and functionally complex entity. We will discuss nutritional functions in our response to Comment 6. We appreciate this helpful comment.

2. While mucous overlies the gastrointestinal epithelium, it is not necessarily a complete barrier to microbes - the degree to which this is true may be different in different regions of the gut, and vary for the types of bacteria studied. Proteobacteria through flagellae (Ann Rev Micro 2011:389), or bacteria using using spirochaetal morphology (e.g. Biophys J 2006:3019), can enter and even transverse the viscous fluids such as those in the mucous layer, as can segmented, filamentous bacteria (SFBs). Scanning EMs of the gut show a variety of interactions where the bacteria sit in the mucous or between the mucous and the surface of epithelium, indeed sometimes bound to the epithelium (e.g. Gut 56:343) and SFBs intimately associate with the epithelium(Nature 520:99). The degree of 'impenetrability' of the mucous layer varies between mammals (Gut 63:281) suggesting that the rules regarding mucous protection of the epithelium are not fully evolved. Indeed, there be development of different mammalian systems, some involving two mucous layers (PNAS 105:15064).

Response: We completely agree with this view that the degree of “*impenetrability*” of the mucus layer varies among mammalian species, gut regions, and bacterial species. We apologize that our original manuscript seemed discordant with this view. Our comparative approach is based on the notion that biological systems can be discriminated into ancestral and derived features when properly set in an evolutionary framework. We showed that the chitin-based system is an ancestral feature of chordates, the loss of which was prerequisite for the initiation of direct interactions between gut microbes and the mucus that overlies the gastrointestinal epithelium. Thus, the aforementioned variation of mucus-microbe interactions, including “*impenetrability*” and “*development of different mammalian systems, some involving two mucous layers,*” would be a derived feature of mammals that occurred after the loss of chitin. As such, comparative analyses using appropriate outgroups advance our understanding of a complex biological trait in a way that is not possible with ingroups *per se* (In our case, outgroups are non-mammalian chordates, and a trait of interest is the mammalian mucus system. Ingroups are mammalian models). We have now explained why we conducted a comparative analysis (P3, line 25–P4, line 4), and we have summarized how current knowledge derived mainly from

mammalian models can be revisited in light of evolution from the chordate ancestor (Supplementary Fig. 1). We then augmented visual explanations about basal chordates (Fig. 1, Supplementary Fig. 2 and Supplementary Movies 1 and 2) and showed how comparative data help us infer diversification of intestinal animal-microbe associations (P10, line 22–P12, line 23). We appreciate this comment, which helped us to reorganize the manuscript.

3. It is not safe to assume that a mesh with pores up to 80-90nm will filter most marine viruses; indeed many viruses are less than 50nm in diameter, especially viruses which are evolved to live in harsher environments and are non-enveloped. The chitinous meshes would be capable of retaining most bacteria of course, assuming they are completely patent.

Response: In accordance with this suggestion, we removed the reference to viruses from the revised manuscript (P5, lines 5–6). We also add explanations that the mesh is usually multi-layered (P5, lines 2–4; P31, lines 14–15), while the negative-staining TEM image (Fig 2c) shows only a single-layered region at the edge of the mesh. This is because multi-layered regions are too thick to transmit a beam of electrons for negative-staining TEM observation, while this observation worked well with single-layered regions. The multi-layered configuration of meshes implies that a functional size for their sieving effect would be much smaller than 65 nm, which is the pore size that we estimated for the single-layered region, though we do not discuss it in the manuscript. We appreciate this comment, which helped us to eliminate an ambiguous passage from the Discussion.

4. Histology is used to demonstrate retention of the macro gut contents. The arrowed spaces between membranes (F1c, SF1g) might be sensitive to fixation artefacts - have these spaces been seen in more conservative histological methods and are they truly bacteria-free?

Response: We had a similar concern about fixation artifacts, especially for Carnoy fixation. Our microscopy was thus carefully conducted by an expert in this technique (S. Kimura). The answer to the question is yes. These spaces were observed using a conservative histological method: Fixation with 2.5% glutaraldehyde, 4% paraformaldehyde, 150 mM NaCl, 100 mM HEPES-KOH (pH 7.2) for 2 h at room temperature; post-fixation with 1% osmium tetroxide, 150 mM NaCl, 100 mM HEPES-KOH (pH 7.2) for 2h on ice; dehydration in a graded ethanol series and substitution with propylene oxide; embedding in epoxy resin, followed by polymerization at 70°C for 16h; ultrathin sectioning (80 nm thickness); and staining with aqueous uranyl acetate and lead citrate. For clarity, we rewrote the whole method section. Methodology for F1c (the revised Fig. 1m) and SF1g (the revised Fig.1n) appear under the subheadings “TEM and negative staining” (P15, lines 12–22) and “Toluidine

blue staining” (P15, line 24–P16, line 3), respectively. The bacteria-free condition was confirmed by PCR amplification of 16S rRNA genes (Fig. 1o). Additionally, cilia sections of fine resolution were similarly observed nearby the gut epithelium, irrespective of the presence or absence of the space (Fig. 1m and Fig. 2i, respectively). Thus, there is no reason to believe that the observed spaces resulted from the fixation process. We appreciate this technical suggestion.

5. Supplementary figure 1 is a key figure and should be in the main body of the paper.

Response: Following this suggestion, we moved the corresponding data to the revised Fig. 1. We thank the reviewer for this positive comment.

6. The study speculates on the purpose of the 'chitinous bags'. While they may have a limited role in protecting the epithelium from e.g. viruses, the presence of anti-bacterial and potentially aggregating proteins within the enclosure may facilitate digestion of the bacteria that enter the organism - their function could be nutritional rather than protective.

Response: It is reasonable to consider that the chitinous bags have nutritional functions, based on the presence of anti-bacterial proteins. At the same time, it is also true that the chitinous bags have protective functions that are essential for survival (e.g., Fig. 2j). We understand that these two functions are not mutually exclusive. Indeed, insect peritrophic matrix, which includes well-studied chitinous bags, has both nutritional and protective functions (Peters, *Peritrophic Membranes*, 1992). Probably, it was a mistake in our original manuscript not to say much about the nutritional functions. Accordingly, we have now explained the nutritional functions in appropriate parts of the revised manuscript. Examples are: “intestinal chitinous membranes, termed PM, have been appreciated for their barrier immunity, nutritional and other physiological functions in invertebrates” (P10, lines 2–3), “In suspension-feeding invertebrates, including basal chordates, in which enzymatic digestion gradually occurs across the semi-permeable chitinous membranes” (P11, lines 9–11), or “chitinous barrier membranes that allow movement of nutrients, but not luminal microbes, onto the ciliated gut epithelium” (P37, lines 4–5). We believe that these additional explanations will provide readers with adequate background information about the nutritional function of the chitinous bags. We highly appreciate this comment.

7. The presence of the pore forming protein (MACPF1) could imply a nutrition-based purpose, rather than an innate defence purpose. The bactericidal assay presented is unconvincing (SF3e) - the protein has no action against E. coli, there is no dose ranging, and evidence that the protein per se is

responsible is not presented. Does the pore forming protein require activation, for example, reduction...? These core issues are not addressed. An antibacterial protein would need to show more efficacy against recognised pathogens of the organisms under investigation for the data to be convincing. Equally, are the bacteria killed or simply in stasis; it is very difficult to tell from the simple data presented. The choice of bacteria, an aerobe which forms spores, is also curious.

Response: Reviewer 1 also raised similar concerns about this experiment and recommended that we remove the corresponding data from the manuscript (Comment 4). We therefore removed it from the revised manuscript. We appreciate this technical advice.

8. Apart from these concerns, the paper looks sound and the biochemical/physical analyses of the various structures is convincing. The paper is by its nature descriptive and somewhat speculative and would be improved by mutagenesis studies which removed one of more actors from the story.

Response: We appreciate this constructive comment. We agree with the importance of mutagenesis studies. We have already started a mutagenesis study using *Ciona*, but it remains technically challenging and will require more time.

9. Minor issues - understanding the microbiota is not simply a question of numbers. Different bacteria interact very differently with the host and the major taxonomic units provide little insight into whether the bacteria have similar or different mucosal interactions - e.g. a flagellated vs non-flagellated bacterium. It is wrong to consider the microbiota of plants or animals as a fixed community. The evolution of the microbiota follows a succession of species, each more fit the incumbent species, reprogrammed by exposure to compounds which grossly affect the organism (e.g. antibiotics). It would be interesting to see what happens when mutations in the various genes encoding proteins active in the chitinous barrier are made.

Response: We agree with this view on the complexity of gut microbiota. We therefore thoroughly revised our manuscript so that it accords with this view. We also agree about the importance of mutation studies, but we need more time for technical reasons. Thanks to the reviewer's comments, we realized that our original manuscript contained ambiguous or speculative points, and we have eliminated them. A clear example is the change of the title, which was originally "Hidden chitin illuminates the origin of mammalian gut microbiota." We believe that the new title "Chitin-based barrier immunity and its loss predated mucus-colonization by indigenous gut microbiota" is much more appropriate for the revised manuscript.

REVIEWERS' COMMENTS:

Reviewer #1 (Remarks to the Author):

The authors have done a very good job in revising the manuscript. I am happy to say I can now fully support the paper for acceptance in Nature Communications. There is only one minor comment: the colour scheme of Fig. 7 is not optimal. There are two yellow blocks with a different meaning. I suppose it is meant that these are different grades of yellow, but, that is not visible. The blue on the yellow looks like green without zooming in.

Reviewer #2 (Remarks to the Author):

The authors have responded adequately and addressed the concerns I raised.

The paper makes an important contribution to evolutionary and ontological understandings of gut function and especially barrier function.